# Heterogeneous Formation of Particulate Nitrate under Ammonium-rich Regimes during the High PM$_{2.5}$ Events in Nanjing, China

**Yu-Chi Lin[1,2,3], Yan-Lin Zhang[1,2,3*], Mei-Yi Fan[1,2,3], Mengying Bao[1,2,3]**

[1.] *Yale-NUIST Center on Atmospheric Environment, International Joint Laboratory on Climate and Environment Change, Nanjing University of Information Science and Technology, Nanjing, 210044, China.*

[2.] *Key Laboratory Meteorological Disaster; Ministry of Education & Collaborative Innovation Center on Forecast and Evaluation of Meteorological Disaster, Nanjing University of Information Science and Technology, Nanjing, 210044, China.*

[3.] *Jiangsu Provincial Key Laboratory of Agricultural Meteorology, College of Applied Meteorology, Nanjing University of Information Science & Technology, Nanjing 210044, China.*

*Corresponded to Yan-Lin Zhang (dryanlinzhang@outlook.com; zhangyanlin@nuist.edu.cn )*

**ABSTRACT**

Particulate nitrate (NO$_3^-$) not only influences regional climates but also contributes to the acidification of terrestrial and aquatic ecosystems. In 2016 and 2017, four intensive online measurements of water-soluble ions in PM$_{2.5}$ were conducted in Nanjing City to investigate the potential formation mechanisms of particulate nitrate. During the sampling periods, NO$_3^-$ was the most predominant species, accounting for 35 % of the total water-soluble inorganic ions, followed by SO$_4^{2-}$ (33 %) and NH$_4^+$ (24 %). Significant enhancements of nitrate aerosols in terms of both absolute concentrations and relative abundances suggested that NO$_3^-$ was a major contributing

species to high-$PM_{2.5}$ events (hourly $PM_{2.5} \geqq 150$ µg m$^{-3}$). High $NO_3^-$
concentrations mainly occurred under $NH_4^+$-rich conditions, implying that the
formation of nitrate aerosols in Nanjing involved $NH_3$. During the high-$PM_{2.5}$ events,
the nitrogen conversion ratios (Fn) were positively correlated with the aerosol liquid
water content (ALWC, $R > 0.72$, $p < 0.05$). Meanwhile, increasing $NO_3^-$
concentrations regularly coincided with increasing ALWC and decreasing Ox (Ox =
$O_3 + NO_2$). These results suggested that the heterogeneous reaction was probably a
major mechanism of nitrate formation during the high-$PM_{2.5}$ events. Moreover, the
average production rate of $NO_3^-$ by heterogeneous processes was estimated to be 12.6
% h$^{-1}$ (4.1 µg m$^{-3}$ h$^{-1}$), which was much higher than that (2.5 % h$^{-1}$, 0.8 µg m$^{-3}$ h$^{-1}$) of
gas-phase reactions. This can also explain the abrupt increases of nitrate
concentrations during the high $PM_{2.5}$ events. Utilization of ISORROPIA II model, we
found that nitrate aerosol formation in Nanjing during the high-$PM_{2.5}$ events was
$HNO_3$-limited. This indicated that control of NOx emissions will be able to efficiently
reduce airborne nitrate concentrations and improve the air quality in this industrial
city.
Keywords: Nitrate aerosols, nitrogen conversion ratios, $NH_4^+$-rich regime, Hydrolysis
of $N_2O_5$, Nitrate production rate

**1.  Introduction**
Due to the rapid growth of industrialization and urbanization, particulate matter
(PM) pollution has become a severe problem in China in recent years (Chan and Yao,
2008; Zhang and Cao, 2015). Fine mode particles ($PM_{2.5}$, with aerodynamic diameters
less than 2.5 µm) exhibit smaller sizes and contain many toxins emitted from
anthropogenic emissions (Huang et al., 2018). $PM_{2.5}$ easily penetrates the upper
respiratory tract and is deposited into the human body, causing serious threats to
human health. Numerous previous studies have proven that people exposed to high
$PM_{2.5}$ concentrations show increased risks of respiratory illness, cardiovascular
diseases and asthma (Brauer et al., 2002; Defino et al., 2005), resulting in an increase
of mortality (Nel, 2005).

Secondary inorganic aerosols (SIA), including sulfate ($SO_4^{2-}$), nitrate ($NO_3^-$) and

ammonium ($NH_4^+$), are major constituents of $PM_{2.5}$, accounting for 25 - 60 % of the
$PM_{2.5}$ mass in urban cities of China (Huang et al.,2014a; Wang et al., 2018; Yang et
al., 2005; Ye et al., 2017; Zhao et al., 2013; Zhou et al., 2018). Among these species,
$SO_4^{2-}$ and $NO_3^-$ are acidic ions which tend to be neutralized by $NH_4^+$. Previously,
many studies suggested that $SO_4^{2-}$ dominated SIA in urban cities of China (Kong et
al., 2014; Tao et al., 2016; Yang et al., 2005; Yao et al., 2002; Zhao et al., 2013). In
recent years, the Chinese government reduced its anthropogenic emissions by 62 %
and 17 % for $SO_2$ and NOx, respectively (Zheng et al., 2018). This revealed that the
reduction efficiency of $SO_2$ emissions were much higher than those of NOx.
Consequently, nitrate has become the dominant species of SIA, especially during PM
haze events (Wang et al., 2018; Wen et al., 2015; Zou et al., 2018).

In the atmosphere, ammonium nitrate ($NH_4NO_3$) is a major form of nitrate

aerosols in fine mode particles. $NH_4NO_3$ is a semi-volatile species which partitions
from the particle phase into the gas phase under high-temperature (T) conditions. It
deliquesces when the ambient relative humidity (RH) is higher than its deliquescence
relative humidity (DRH, nearly 62 % RH at atmospheric standard condition). To
produce $NH_4NO_3$, nitrogen oxides ($NO_x$) and ammonia ($NH_3$) undergo a series of
chemical reactions. $NO_x$ mostly emits as fresh NO, which is subsequently oxidized to
$NO_2$ and reacts with hydroxyl (OH) radicals to generate nitric acid ($HNO_3$), and then
$HNO_3$ reacts with $NH_3$ to yield $NH_4NO_3$ particles as listed in R1 and R2 (Calvert and
Stockwell, 1983). Particulate $NH_4NO_3$ formation rate is profoundly dependent on the
ambient T and RH since both parameters influence the equilibrium constant of $NO_3^-$
and $NH_4^+$ between the particle and gas phases, as listed in R2 (Lin and Cheng, 2007).
$$NO_{2(g)} + OH_{(g)} \rightarrow HNO_{3(g)} \qquad k_1 \qquad (R1)$$
$$HNO_{3(g)} + NH_{3(g)} \rightarrow NH_4NO_{3(s,\,aq)} \qquad k_2 \qquad (R2)$$
Here, $k_1$ and $k_2$ are the reaction rate and equilibrium constant of R1 and R2,
respectively. The equilibrium constant $k_2$ can be expressed as the product of $HNO_3$
and $NH_3$.
Heterogeneous reactions have been considered an important mechanism of nitrate
formation during nighttime. As listed in R3, liquid $HNO_3$ is produced by the
hydrolysis of dinitrogen pentoxide ($N_2O_5$) on aerosol surfaces (Brown & Stutz, 2012;
Chang et al., 2011; Mental et al., 1999; Wahner et al., 1998). Liquid $HNO_3$ can be
neutralized by $NH_4^+$, which is produced from the conversion of gaseous $NH_3$. Nitrate
aerosols yielded from both R2 and R3 require $NH_3$, and therefore these processes of
$NO_3^-$ formation occur under $NH_4$ -rich conditions. Sometimes, there is not enough
$NH_3$ ($NH_4^+$) to react (to be neutralized) with $HNO_3$ ($NO_3^-$) after complete
neutralization by $H_2SO_4$. Under this condition, $HNO_3$ tends to react (or to be
neutralized) with other alkaline species such as Ca-rich dust ($CaCO_3$), and
subsequently, nitrate aerosol is produced under a $NH_4^+$-poor regime (Goodman et al.,

2000).

$$N_2O_{5(g)} + H_2O_{(l)} \rightarrow 2HNO_{3(aq)} \qquad (R3)$$

The Yangtze River Delta (YRD) region is one of the well-known polluted areas
in China (Zhang and Cao, 2015). Different from the case of dramatic elevated sulfate
aerosol levels in Beijing (Wang et al., 2016), nitrate aerosols seemed to be a major
contributing species during haze days in the YRD region (Wang et al., 2015; Wang et
al., 2018). The formation mechanisms of nitrate in Nanjing have not yet been well
understood, especially during high PM events. In this study, four intensive online
measurements of water-soluble ions in $PM_{2.5}$ were conducted in Nanjing City in 2016
and 2017. The data provided information on the hourly evolution of water-soluble
inorganic ions (WSIIs) in the industrial city. The $NO_3^-$ distributions under different
$NH_4^+$ regimes ($NH_4^+$-poor and $NH_4^+$-rich conditions) were also discussed. Finally, we
investigated the potential formation mechanisms of nitrate aerosols and their
production rates during high-$PM_{2.5}$ events based on the online measurements.

**2.   Methodology**
**2.1 Sampling site**

Particulate WSIIs and inorganic gases were continuously monitored at Nanjing

University of Information Science and Technology (NUIST) located in the northern
part of Nanjing City (see Figure S1). In addition to vehicle emissions, petroleum
chemical refineries and steel manufacturing plants situated in the northeast and east
direction at a distance of approximately 5 km are also major anthropogenic emissions
near the receptor site. Four intensive campaigns were conducted from March 2016 to
August 2017. During each experiment, the hourly concentrations of WSIIs in $PM_{2.5}$
and inorganic gases were continuously observed. Meanwhile, the hourly $PM_{2.5}$ mass,
$NO_2$ and $O_3$ concentrations along with the ambient T and RH were acquired from the
Pukou air quality monitoring station which is located to the southwest of the receptor
site.

**2.2 Instruments**

To monitor the hourly concentrations of WSIIs ($Cl^-$, $NO_3^-$, $SO_4^{2-}$, $Na^+$, $NH_4^+$, $K^+$,

$Mg^{2+}$ and $Ca^{2+}$), an online Monitor for Aerosols and Gases (MAGAR, Applikon-ENC,
The Netherlands) instrument with a $PM_{2.5}$ inlet was employed. Using this instrument,
the WSIIs in $PM_{2.5}$ were collected by a stream jet aerosol collector, while acidic (HCl,
HONO, $HNO_3$ and $SO_2$) and basic gases ($NH_3$) were dissolved in a hydrogen peroxide
solution on a wet rotation denuder (ten Brink et al., 2007; Griffith, et al., 2015). The
liquid samples were then collected with syringe pumps and analyzed by ion
chromatography (IC). Before each campaign, a seven-point calibration curve of each
species was made, and an internal standard solution (LiBr) was used to check
instrumental drifts. The method detection limits (MDLs) of $Cl^-$, $NO_3^-$, $SO_4^{2-}$, $Na^+$,
$NH_4^+$, $K^+$, $Mg^{2+}$ and $Ca^{2+}$ were, 0.01, 0.04, 0.06, 0.05, 0.05, 0.07, 0.05 and 0.11 $\mu g\ m^-$
$^3$, respectively. For gases, the MDLs were 0.07, 0.09, 0.06, 0.02 and 0.08 $\mu g\ m^{-3}$ for
HCl, HONO, $HNO_3$, $SO_2$ and $NH_3$, respectively.

**2.3 ISORROPIA-II model**
In this work, we used the ISORROPIA-II model to calculate the aerosol liquid
water content (ALWC). ISORROPIA II is a thermodynamic equilibrium model which
is built based on the $Na^+$ - $Cl^-$ - $Ca^{2+}$ - $K^+$ - $Mg^{2+}$ - $SO_4^{2-}$ - $NH_4^+$ - $NO_3^-$ - $H_2O$ aerosol
system (Fountoukis & Nenes, 2007). This model has been successfully used to
estimate the liquid water content in aerosols with uncertainty of ~ 20 % compared to
the observed ALWC (Bian et al., 2014; Guo et al., 2015; Liu et al., 2017). This
underestimation might be due to the missed species in ISORROPIA II, organic
aerosols, which contributed approximately 27 % to total ALWC (Bougiatioti et al.,
2007). Here, the model was computed as a "forward problem", in which the quantities
of aerosol- and gas-phase compositions along with the T and RH were well known.
Additionally, the modeled values were determined using the "metastable" mode,
which indicated that the aerosol compositions were assumed to be composed of an
aqueous solution (Liu et al., 2017). The details of this model can be found elsewhere
(Fountoukis and Nenes, 2007). In this work, the observed concentrations of total
nitrate ($HNO_3 + NO_3^-$), total ammonium ($NH_3 + NH_4^+$), total chloride ($HCl + Cl-$),
$SO_4^{2-}$, $Na^+$, $K^+$, $Mg^{2+}$ and $Ca^{2+}$ along with measured ambient T and RH served as
input of ISORROPIA II model.

**2.4 Potential source contribution function**

Potential source contribution function (PSCF) is a method to identify the

potential source regions of air pollutants. It has also been widely used to differentiate
local emission from long-range transported pollution (Zhang et al., 2013; Hui et al.,
2018) based on the trajectory analysis calculated from GDAS (Global Data
Assimilation System), which processed by the National Centers for Environmental
Prediction (NCEP). The zone of concern is divided into $i \times j$ small equal grid cells and
then PSCF in the *i-jth* cell (*PSCF_{ij}*) can be defined as (Polissar et al., 1999):
$$PSCF_{ij} = \frac{m_{ij}}{n_{ij}} \tag{1}$$
where $m_{ij}$ is the number of "high nitrate pollution" trajectory endpoints in the *i-jth* cell
and $n_{ij}$ is the total number of trajectory endpoints fallen into the *i-jth* cell. In this
study, the 80th percentile value of nitrate concentration was treated as "high nitrate
pollution" threshold. To reduce the uncertainty caused by the small values of $n_{ij}$, the
weighting function of $W_{ij}$ has to be considered (Polissar et al., 1999):

$$W_{ij} = \begin{cases} 1.00; & 80 < n_{ij} \\ 0.70; & 20 < n_{ij} \leqq 80 \\ 0.42; & 10 < n_{ij} \leqq 20 \\ 0.05; & n_{ij} \leqq 10 \end{cases}$$

In this study, the domain of the study area was in a range of 20-55 °N and 105-135 °E;
the resolution of grid cell was 0.5°×0.5°.

**3.   Results and discussion**
**3.1 Overview of water-soluble inorganic ions**
Four intensive online measurements of WSIIs in $PM_{2.5}$ were carried out in
Nanjing City from March 2016 to August 2017. Figure 1a plots the time series of the
hourly $PM_{2.5}$ mass concentrations during the sampling periods. As seen, the hourly
$PM_{2.5}$ mass concentrations varied from 5 to 252 $\mu g\ m^{-3}$ with a mean value of $58 \pm 35$
$\mu g\ m^{-3}$. Compared with the 24-hour guideline (25 $\mu g\ m^{-3}$) suggested by the World
Health Organization (WHO), our average $PM_{2.5}$ concentration (58 $\mu g\ m^{-3}$) was 2.3
times higher. This indicated that PM pollution in Nanjing City was a serious problem.
During the campaigns, several high-$PM_{2.5}$ events with hourly $PM_{2.5}$ concentrations of
higher than 150 $\mu g\ m^{-3}$ were observed in the springtime and wintertime. These high
$PM_{2.5}$ levels lasted for more than 3 hours, with obviously elevated $NO_3^-$. The details
of nitrate formation during the high-$PM_{2.5}$ hours will be discussed in the following
sections.
Figure 1b shows time series of the hourly concentrations of SIA species,
including $SO_4^{2-}$, $NO_3^-$ and $NH_4^+$. The lack of data from March 7 to 14, 2016 was due
to a malfunction of the MARGA instrument. During the sampling periods, the $NO_3^-$
concentrations varied from 0.1 to 85.1 $\mu g\ m^{-3}$ with a mean value of $16.7 \pm 12.8$ $\mu g\ m^{-}$
$^{3}$. The $SO_4^{2-}$ concentrations ranged from 1.7 to 96.2 $\mu g\ m^{-3}$ and averaged $14.9 \pm 9.1$
$\mu g\ m^{-3}$. The $NH_4^+$ concentrations fluctuated between 0.8 and 44.9 $\mu g\ m^{-3}$ with a mean
value of $10.7 \pm 6.7$ $\mu g\ m^{-3}$. On average, SIA accounted for 91 % of the total water-
soluble inorganic ions (TWSIIs) during the entirety of the sampling periods (see
Figure 2a). Among these species, $NO_3^-$ accounted for 35 % of the TWSIIs, followed
by $SO_4^{2-}$ (33 %) and $NH_4^+$ (24 %). The abundances of other ions, including $Cl^-$, $K^+$,
$Ca^{2+}$, $Na^+$ and $Mg^{2+}$, were 5, 2, 1, 0.7 and 0.3 %, respectively. Figure S2 shows the
scatter plot of the equivalent concentrations of the cations ($Na^+$, $NH_4^+$, $K^+$, $Mg^{2+}$ and
$Ca^{2+}$) and anions ($Cl^-$, $SO_4^{2-}$ and $NH_4^+$). As seen, good correlations (R = 0.98 -0.99,
with a significance level $p<0.05$) between cations and anions were found during the
various sampling periods. The ratio of cation-to-anion was very close to 1.0 during
each season, reflecting an ionic balance. This also indicated that our data exhibited
good quality and was able to be used for the further analysis of scientific issues.

All SIA species exhibited similar seasonal patterns, with lower concentrations in

the summer, especially for $NO_3^-$. The average concentrations of nitrate were 6.7 and
5.7 $\mu g\ m^{-3}$ in the summertime of 2016 and 2017, respectively (see Figure S3). These
values were much lower than those observed during other seasons. The local
meteorological conditions, which were favorable for the dilution of air pollution, were
one of the reasons for the declined $NO_3^-$ concentrations during the hot seasons (Zhang
and Cao, 2015). Another important reason for this effect was attributed to the
formation process of $PM_{2.5}$ nitrate, which is very sensitive to the ambient T and RH
(Lin and Cheng, 2007). Figure S4a depicts the theoretical equilibrium constants of
partitioned $NO_3^-$ and $NH_4^+$ between the particle and gas phase ($HNO_{3(g)}$ + $NH_{3(g)}$ →
$NH_4NO_{3(s,\ aq)}$ as seen in R2) under different T and RH conditions. The details of
calculation approach of the theoretical equilibrium constants are described in
*Supplementary S1*. Note that the *Y-axis* is presented on a log scale. The theoretical
equilibrium constants increased exponentially with increasing ambient temperature
but decreased with increasing RH. This indicated that $NH_4NO_3$ would be partitioned
into the gas phase due to high equilibrium constants under high-temperature and low-
RH conditions. Figure S4b illustrates the time series of the theoretical and observed
equilibrium constants during the sampling periods. As can be seen, most of the
observed equilibrium constants were higher than the theoretical ones, suggesting that
NH$_4$NO$_3$ aerosols were produced in Nanjing during the sampling periods. Obviously,
higher theoretical and lower observed equilibrium constants were found during the
summer. This suggested that more NO$_3^-$ and NH$_4^+$ would tend to be partitioned into
the gas phase, resulting in lower particulate nitrate concentrations during hot seasons
(Lin and Cheng, 2007).

Apart from seasonal variations, pronounced diurnal patterns were also found for

SIA species (see Figure 3). NO$_3^-$ exhibited similar diel cycles during different seasons,
with higher concentrations in the early morning (3 a.m. - 7 a.m.) and lower levels
between 2 p.m. and 5 p.m. The high nitrate concentrations in the early morning might
be caused by the nitrate formation via heterogeneous reaction in the dark, and gas-
phase oxidation after sunrise and the subsequent condensation on pre-existing
particles before the temperature increased and RH decreased afterwards. Moreover,
the lower planet boundary layer (PBL) might be another reason for enhanced nitrate
in the early morning. However, the lower concentrations of nitrate during the daytime
might be attributed to the higher PBL, and high temperatures, which inhibited the
build-up of nitrate, especially during the summertime. In terms of sulfate, higher
concentrations were observed between 6 am. and 1 p.m., indicating that the formation
rate of sulfate was higher than the removal/dilution rate, leading to an increase of the
sulfate concentration during the daytime. The diurnal patterns of NH$_4^+$ mimicked
those of NO$_3^-$, showing lower concentrations during the daytime. This was explained
by the drastic decrease of particulate NH$_4$NO$_3$ concentrations under high temperatures
and low relative humidity, resulting in lower NH$_4^+$ levels during the daytime.

**3.2 Enhancements of nitrate at high PM$_{2.5}$ levels**

Figure S5 shows the scatter plots of NO$_3^-$, SO$_4^{2-}$ and NH$_4^+$ against PM$_{2.5}$. As

seen, the slopes of NO$_3^-$ (NO$_3^-$ vs. PM$_{2.5}$ mass), SO$_4^{2-}$ and NH$_4^+$ were 0.30, 0.24 and
0.19, respectively. This suggested that the increasing rate of $NO_3^-$ during the high-
$PM_{2.5}$ events was higher than those of other SIA species. At high $PM_{2.5}$ levels ($PM_{2.5} \geq$
150 μg/m³), $NO_3^-$, $SO_4^{2-}$ and $NH_4^+$ contributed 39, 28 and 24 % of the TWSIIs,
respectively (Figure 2b). However, the relative abundances of $NO_3^-$, $SO_4^{2-}$ and $NH_4^+$
during low $PM_{2.5}$ concentrations (hourly $PM_{2.5} < 35$ μg/m³, see Figure 2c) were 29, 37
and 23 %, respectively. In recent years, dramatically enhanced amounts of nitrate
aerosols during high-PM events have been observed at many urban sites in China
(Wen et al., 2015; Wang et al., 2017; 2018; Zou et al., 2018). For instance, Zou et al.
(2018) found that the nitrate concentrations during the occurrence of polluted air in
Beijing and Tianjin were almost 14 times higher than those on relatively clean days
($PM_{2.5} < 75$ μg/m³), and the enhancement ratio of nitrate was much higher than that
(5.3) of sulfate. Wang et al. (2018) noted that the enhancement ratio of $NO_3^-$ (~6)
between haze and clear days in Ningbo of the YRD region was much higher than that
of $SO_4^{2-}$ (~3). These findings suggested that $NO_3^-$ was a major contributing species to
fine particles during haze days since its increasing ratio between haze and non-haze
days was much higher than those of other SIA species, such as sulfate and
ammonium.

**3.3 PSCF result of high nitrate pollution**

During the high $PM_{2.5}$ pollution, significant enhanced nitrate aerosols in terms

of both absolute concentration and relative abundance to TWSIIs were found. Next,
we tried to use PSCF analysis to identify whether local emission or long-range
transported pollution was the major source of high nitrate concentrations at the
receptor site. In this work, the 80th percentile values of nitrate concentration was
selected as "high nitrate pollution" threshold for PSCF analysis. Figure 4 plots the
PSCF result of high nitrate pollution in Nanjing during the sampling periods. The
region corresponding to high PSCF value grid is a potential source region of nitrate
aerosols. As can be seen, the areas with high PSCF value (>0.8) were regularly local
areas surrounding by Nanjing while PSCF values from other long-distance areas were
lower than 0.2. This suggested that $NO_3^-$ aerosols in Nanjing during the high nitrate
pollution were likely from local emissions rather than long-range transported sources.

**3.4 Nitrate formation under different ammonium regimes**
Ammonium is a major species that neutralizes particulate $SO_4^{2-}$ and $NO_3^-$. In the
atmosphere, $SO_4^{2-}$ competes with $NO_3^-$ for $NH_4^+$ during their formation processes, and
therefore, the relationship between the molar ratios of $NO_3^-/SO_4^{2-}$ and $NH_4^+/SO_4^{2-}$ can
give us a hint for understanding the formation of $NO_3^-$ under different ammonium
regimes (Pathak et al., 2009; He et al., 2012; Tao et al., 2016). In an ammonium-rich
regime, the $HNO_3$ produced by both gas oxidation and heterogeneous process reacts
(or neutralizes) with "excess-ammonium" (excess-$NH_4^+$) at a $NH_4^+/SO_4^{2-}$ molar ratio
> 2 (theoretical value in an $NH_4^+$-rich regime) when sulfate is completely neutralized
by $NH_4^+$ to form $(NH_4)_2SO_4$ (Squizzato et al., 2013; Ye et al., 2011). In contrast,
nitrate can be found under ammonium-poor conditions with a theoretical $NH_4^+/SO_4^{2-}$
value that should be less than 2 (Pathak et al., 2009). Under $NH_4^+$-poor conditions,
$HNO_3$ reacts with other cations, such as the calcium carbonate, frequently found in
natural dust.
Figure 5 shows the scatter plot of the molar ratios of $NO_3^-/SO_4^{2-}$ against
$NH_4^+/SO_4^{2-}$. It is found that good correlations existed between $NO_3^-/SO_4^{2-}$ and
$NH_4^+/SO_4^{2-}$ under $NH_4^+$-rich regimes, with a coefficient of determination ($R^2$) of 0.84
- 0.94 in the different seasons (see in Table 1). Utilizing the linear regression model,
we suggested that nitrate aerosols (in $NH_4^+$-rich regimes) began to form when the
$NH_4^+/SO_4^{2-}$ molar ratios exceeded the criterion values of 1.7-2.0 during the different
seasons (Table 1). The criterion value can be calculated as absolute value of
"intercept" dividing by slope in each linear regression model (He et al., 2012). The
criterion values below 2 suggested that part of the sulfate might have existed in other
forms, such as ammonium bisulfate. On the other hand, under ammonium-rich
conditions, nitrate concentrations should be positively proportional to "excess-$NH_4^+$"
concentrations, a relationship which was defined as [excess-$NH_4^+$] = ($NH_4^+/SO_4^{2-}$ -
criterion value) × [$SO_4.^{2-}$] (Pathak et al., 2009) (sulfate is in the units of nmol m$^{-3}$
here). The criterion values were acquired from the regression models, as listed in
Table 1. The results revealed that the excess-$NH_4^+$ concentrations varied from -283 to
1422 nmol m$^{-3}$ (see Figure 6), and only 1 % of data showed deficit-$NH_4^+$ conditions,
reflecting that $NO_3^-$ formation in Nanjing occurred primarily under the $NH_4^+$-rich
conditions. Moreover, the excess-$NH_4^+$ had apparent diurnal cycles, with higher
concentrations in the early morning and lower concentrations at midday and in the
early afternoon (see Figure 3, where we converted the units from nmol m$^{-3}$ to μg m$^{-3}$).
The diurnal patterns of $NO_3^-$ mimicked those of the excess-$NH_4^+$. This also suggested
that particulate $NO_3^-$ formation occurred mainly under $NH_4^+$-rich conditions. Figure 6
illustrates the relationship between the nitrate and excess-$NH_4^+$ molar concentrations
during the sampling periods. The nitrate molar concentrations correlated linearly with
the excess-$NH_4^+$ molar concentrations with a slope of approximately 1.0, which was
consistent with the molar ratio of reaction between $HNO_3$ and $NH_3$. Interestingly,
some scattered points were found in high ammonium concentrations (excess-$NH_4^+$ ≧
900 nmol m$^{-3}$ ~ 16.2 μg m$^{-3}$), implying that residual $NH_4^+$ might be presented in
another form such as $NH_4Cl$ under high-$NH_4^+$ conditions. On the contrary, $NO_3^-$
aerosols can be produced without involving $NH_3$; therefore, $NO_3^-$ did not correlate
well with the excess $NH_4^+$ under a $NH_4^+$-poor regime.

In this study, high nitrate concentrations were always found under $NH_4^+$-rich

regimes, elucidating that nitrate production during high PM levels in Nanjing had to
be involved with $NH_3$ or $NH_4^+$. Figure 6 also shows the nitrate concentrations against
the excess-$NH_4^+$ observed in various cities of China during the summertime (Pathak et
al., 2009; Griffith et al., 2015). In Beijing and Shanghai, high nitrate concentrations
during the summertime were found under $NH_4^+$-deficient conditions, which was very
different from the findings of this work. In these studies (Pathak et al., 2009; Griffith
et al., 2015), the high nitrate concentrations associated with $NH_4^+$-poor conditions
might be due to the lower excess-$NH_4^+$ concentrations under high-$SO_4^{2-}$ conditions at
that time since the strict control of $SO_2$ emissions by the Chinese government started
in 2010 (Zheng et al., 2018). In recent years, the reduction of anthropogenic $SO_2$
emissions decreased the airborne $SO_4^{2-}$ concentrations, resulting in more excess-$NH_4^+$
and leading to nitrate aerosol formation under $NH_4^+$-rich regimes. This argument can
be supported by the recent results shown in Figure S6, in which high nitrate
concentrations in Beijing were always found under $NH_4^+$-rich regimes.

**3.5 Nitrate formation mechanism during high-PM$_{2.5}$ episodes**
In this section, we attempted to explore the formation mechanisms of nitrate
aerosols during high $PM_{2.5}$ levels. Here, nitrogen conversion ratio (Fn) was used to
evaluate the conversion capability of $NO_2$ to total nitrate (TN, TN=$HNO_3 + NO_3^-$),
and it can be defined as (Khoder, 2002; Lin et al., 2006):

$$F_n = \frac{GNO_3^- + PNO_3^-}{GNO_3^- + PNO_3^- + NO_2}$$

(1)

where $GNO_3^-$ and $PNO_3^-$ represent the $NO_2$ concentrations in nitric acid and
particulate nitrate, respectively, with the units of $\mu g\ m^{-3}$. The results showed that the
Fn values during the sampling periods varied from 0.01 to 0.57 with a mean value of
$0.14 \pm 0.09$ (see Figure 1e). This value was comparable to that (0.17) in Taichung,
Taiwan, where both gas-oxidation and heterogeneous reaction were the dominant
formation mechanisms of atmospheric $HNO_3$ (or $NO_3^-$) (Lin et al., 2006). However,
our Fn value was 2.3 time higher than that (0.06) in Dokki, Egypt (Khoder, 2002).
The reason of significant discrepancy of Fn between this work and that in Dokki was
not clearly understood, but it might be attributed to different formation processes of
$HNO_3$. In Dokki, gas-phase oxidation was the dominant pathway of $HNO_3$ production
while heterogeneous process (R3) played an important role in $HNO_3$ formation in
addition to gas-phase oxidation in Nanjing, especially during the high-$PM_{2.5}$ events
(discussed later). The reaction rate of $HNO_3$ by heterogeneous process was much
higher than that by gas-phase oxidation (Calvert and Stockwell, 1983) and therefore,
the Fn value was much higher in this study. On the other hand, Fn displayed
significant diurnal cycles, with the highest value in the early morning (see in Figure
3). This elevated Fn coincided with increasing ALWC, suggesting heterogeneous
reaction since ALWC is one of the key parameters which favors the transformation of
$N_2O_5$ to liquid $HNO_3$ in this process (also indicated that nitrate formation was
associated with heterogeneous process). On the contrary, a second peak of Fn was
found in the early afternoon when Ox (Ox = $NO_2$ + $O_3$, an index of the oxidation
capacity) concentrations increased, but ALWC decreased. This suggested that the
$HNO_3$ formation might be mainly associated with the gas-phase reaction of $NO_2$ +
OH during the daytime; also reflected that nitrate formation was via gas-phase
oxidation.

Assuming that long-range transported nitrate can be neglected in this study (in

section 3.3), we attempted to analyze the correlations of Fn vs. OH and Fn vs. ALWC
to investigate whether gas-phase oxidation or heterogeneous reactions might be the
dominant mechanism of nitrate production. In this work, the OH radical
concentrations were not measured; hence, we used $O_X$ as a proxy of OH. The ALWC
was acquired by computing the ISOPROPIA II model as described in section 2.3.
Figure 7 illustrates the scatter plots of Fn against Ox and ALWC in both daytime and
nighttime aerosol samples during the high-$PM_{2.5}$ events. Fn correlated well with the
ALWC, with a correlation coefficient (R) of 0.72 and 0.76 ($p < 0.05$) at daytime and
nighttime samples, respectively. However, a poor correlation was found between Fn
and Ox (R was 0.17 and 0.52 for the daytime and nighttime samples, $p>0.05$). This
implied that nitrate formation during the high-$PM_{2.5}$ events in Nanjing was likely
attributed to heterogeneous reactions. This result was consistent with recent
conclusions reached by oxygen isotope techniques, in which the hydrolysis of $N_2O_5$ in
preexisting aerosols was found to be a major mechanism of $NO_3^-$ formation (Chang et
al., 2018).

**3.6 Case study and production rate of $NO_3^-$ during $PM_{2.5}$ episodes**

Figure 8 shows several high-$PM_{2.5}$ events observed from March 3 to 6 in 2016.

In case I, the high $PM_{2.5}$ concentrations started at 6 p.m. on March 3 and ended at 3
a.m. on March 4. During this event, the $SO_4^{2-}$ and $NH_4^+$ concentrations remained at
almost constant levels, but the $NO_3^-$ concentrations revealed a slight enhancement. In
the early morning of March 4, the $NO_3^-$ concentrations increased from 39.4 to 47.8 μg
$m^{-3}$ within 4 hours, resulting in a nitrate production rate of 2.3 μg $m^{-3}$ $h^{-1}$ (~5.5 % $h^{-1}$,
the calculation of $NO_3^-$ production rate can be seen in the *Supplementary S2*). In case
II, high $PM_{2.5}$ concentrations were observed from 8. a.m. to 2. p.m. on March 4. The
$NO_3^-$ concentrations were much higher than those of $SO_4^{2-}$, indicating nitrate-
dominated aerosols. In this case, the $NO_3^-$ concentrations increased from 38.1 to 51.2
μg $m^{-3}$ within 6 hours, suggesting that the increasing rate of $NO_3^-$ was 1.0 μg $m^{-3}$ $h^{-1}$
(2.4 % $h^{-1}$). Since the high $NO_3^-$ concentrations occurred under high-Ox and low-
ALWC conditions, this suggested that the gas-phase reaction of $NO_2$ + OH might be
the dominant source of $NO_3^-$ production in this event. In case III, a rapid growth of the
$PM_{2.5}$ mass was found around midnight, along with a dramatic increase of $NO_3^-$
concentrations from 11 p.m. on March 4 (31.0 $\mu$g m$^{-3}$) and maximizing at 1 a.m. the
next day (64.5 $\mu$g m$^{-3}$). The increasing rate of $NO_3^-$ was estimated to be 11.4 $\mu$g m$^{-3}$ h$^-$
$^1$ (~26.7 % h$^{-1}$), which was much higher than those in case I and II. The high-nitrate
event was found under increasing ALWC and decreasing Ox concentration conditions,
suggesting that nitrate production occurred through heterogeneous processes. In case
IV, the enhancements of all SIA species coincided with increasing ALWC and
declining Ox concentrations. Again, the enhancement of nitrate was likely attributed
to heterogeneous reactions rather than to gas-phase processes. In these events, the
$NO_3^-$ production rate was estimated to be 5.0 $\mu$g m$^{-3}$ h$^{-1}$ (~ 15.4 % h$^{-1}$).
Through the sampling periods, a total of twelve high $PM_{2.5}$ events were found, and the
$NO_3^-$ concentrations increased significantly during all the episodes (see in Table S1).
Seven episodes suggested that heterogeneous processes ($N_2O_5$ + $H_2O$) might be a
major pathway for nitrate formation since elevated $NO_3^-$ levels coincided with
increasing AWLC and decreasing Ox (or Ox remaining at a constant level). Among
these heterogeneous process events, five cases (Case III, Case IX, Case X, Case XI
and Case XII in Table S1) were observed during the nighttime (5 p.m. – 6 a.m. on the
next day). This suggested that approximately 70 % heterogeneous reaction of nitrate
production was observed in the dark. In these events, the average $NO_3^-$ growth rate
was 12.6 ± 7.3 % h$^{-1}$ (4.1 ± 3.6 $\mu$g m$^{-3}$ h$^{-1}$). This value was in agreement with those in
the literatures which the production rate of nitrate via heterogeneous reaction were
14.3 % h$^{-1}$ by both field measurements and laboratory works (Calvert and  Stockwell,
1983; Pathak et al., 2011). On the contrary, $NO_3^-$ concentrations rose with increasing
Ox and decreasing ALWC in two $PM_{2.5}$ episodes, indicating gas-phase processes
($NO_2$ + OH). As listed in Table S1, these gas-phase reaction cases occurred mainly
during the daytime. The average production rate of $NO_3^-$ in the gas-oxidation reaction
cases averaged $2.5 \pm 0.1$ % $h^{-1}$ ($0.8 \pm 0.3$ $\mu g$ $m^{-3}$ $h^{-1}$), which was in line with that (2.4
% $h^{-1}$) in the subtropical polluted urban site that nitrate aerosols were mainly from
gas-oxidation process (Lin et al., 2007). Moreover, we also found some cases in
which the elevated $NO_3^-$ might have been from both gas-phase and heterogeneous
reactions, and the corresponding $NO_3^-$ growth rate was approximately $7.5 \pm 3.0$ % $h^-$
$^1$($2.5 \pm 0.2$ $\mu g$ $m^{-3}$ $h^{-1}$). In conclusion, enhancements of $NO_3^-$ in Nanjing usually
occurred under increased ALWC and decreased Ox conditions, indicating that
heterogeneous reactions provided the dominant pathway of nitrate formation during
the $PM_{2.5}$ episodes. Moreover, the average growth rate of $NO_3^-$ (12.6 % $h^{-1}$) by
heterogeneous processes was 5 times higher than that (2.5 % $h^{-1}$) of gas-phase
reactions. This might explain the abrupt increase of nitrate concentrations during the
high $PM_{2.5}$ events.

**3.7 $HNO_3$/$NH_3$ limitation of nitrate aerosol formation**

In Nanjing, high nitrate concentrations occurred mainly under $NH_4^+$-rich

regimes, indicating the involvement of atmospheric $NH_3$. This also demonstrated that
both $HNO_3$ and $NH_3$ were crucial precursors for particulate nitrate formation. In this
section, we attempted to discuss whether $HNO_3$ or $NH_3$ was the limited factor for
nitrate formation in Nanjing during the high-$PM_{2.5}$ events. ISORROIPA II model is
capable of predicting concentrations of particulate ions in addition to ALWC under
thermodynamic equilibrium between gas- and aerosol-phase of these ions (Tang et al.,
2016). In section 3.5, we used this model to estimate ALWC. Indeed, the output data
also included concentrations of ionic species. Figure S7 illustrates the scatter plots of
modeled results against observations of $NO_3^-$, $SO_4^{2-}$ and $NH_4^+$ in Nanjing during the
sampling periods. Good correlations were found between modeled results and
observations ($R^2$=0.97-0.99 with all slopes of approximately 1.0), suggesting that
ISORROPIA II had a good performance in prediction of SIA species. As a result, we
can use ISORROPIA II model to test sensitivity of $HNO_3$ and $NH_3$ to particulate
nitrate concentrations (Guo et al., 2018).
Figure 7 shows the contour plot of the simulated nitrate concentrations depending
on the various total nitrate (TN) and total ammonium (TA, TA=$NH_3$ + $NH_4^+$) levels
under thermodynamic equilibrium conditions computed by ISORROPIA II model.
The details of considered chemical reactions in ISORROPIA II model can be seen
elsewhere (Fountoukis & Nenes, 2007). Here, sulfate concentrations were assumed to
be 10 and 60 μg m$^{-3}$ for the tests of different sulfate conditions. The average
concentrations of total chloride (HCl + Cl$^-$, 1.3 μg m$^{-3}$), Na$^+$ (0.2 μg m$^{-3}$), K$^+$ (0.8   μg
m$^{-3}$), Mg$^{2+}$ (0.1 μg m$^{-3}$) and Ca$^{2+}$ (0.5 μg m$^{-3}$) along with ambient T (20 °C) and RH
(62 %) at the receptor site during the sampling period served as input data in this
model. The results showed that the lower simulated $NO_3^-$ concentrations was found in
the higher $SO_4^{2-}$ case under the same TN and TA levels. This was attributed to less
$NH_4NO_3$ formation under higher $SO_4^{2-}$ conditions since $SO_4^{2-}$ would compete with
$NO_3^-$ for $NH_4^+$.
According to the simulated results, we can roughly split the plots into two parts:
one is $HNO_3$-limited area (right), and another is $NH_3$-limited region (left). The
observed TN and TA concentrations (pink circles) in Nanjing are also plotted in this
figure. Most of the observed data sets were mainly affected by TN under a low-$SO_4^{2-}$
case. Under a high-$SO_4^{2-}$ condition, the observed data fell into TA-limited under a
low-TN and -TA regime, but fell into TN-limited in high-TA and-TN regimes. During
the sampling period, high nitrate concentrations always occurred under the high TN
and TA conditions, highlighting that nitrate aerosol production in Nanjing during the
high $PM_{2.5}$ levels was mainly control by $HNO_3$. Therefore, control of NOx emissions,
which reduced $HNO_3$ concentrations, might be an important way to decrease airborne
nitrate concentrations and ameliorate the air quality in Nanjing.

**4.    Conclusion and remarks**
Four intensive online measurements of water-soluble ions in $PM_{2.5}$ were carried
out in Nanjing City in 2016 and 2017 to realize the evolutions of SIA and the potential
formation mechanisms of particulate nitrate. During the sampling periods, the average
concentrations of $NO_3^-$, $SO_4^{2-}$ and $NH_4^+$ were 16.7, 14.9 and 10.7 $\mu g\ m^{-3}$, respectively.
This indicated that $NO_3^-$ dominated the SIA. Significant seasonal variations and
diurnal cycles were found for all SIA species. The low $NO_3^-$ concentrations observed
during the summer daytime could be attributed to the enhanced theoretical and
declined observed equilibrium constants of $NO_3^-$ and $NH_4^+$ between gas- and particle-
phase. Obvious enhancements of $NO_3^-$ were found in terms of both absolute
concentrations and relative abundances during the $PM_{2.5}$ episodes, indicating that
$NO_3^-$ was a major contributing species to $PM_{2.5}$. Different from the results obtained in
Beijing and Shanghai, high nitrate concentrations always occurred under $NH_4^+$-rich
regimes. The nitrogen conversion ratio, Fn, correlated well with the ALWC but not
with Ox during high-$PM_{2.5}$ episodes. These findings indicated that $NO_3^-$ aerosols at
the receptor site were mainly produced by heterogeneous reactions ($N_2O_5 + H_2O$) with
the involvement of $NH_3$. The average production rate of $NO_3^-$ from heterogeneous
reactions was estimated to be 12.6 % $h^{-1}$, which was 5 time higher than that of gas-
phase reactions. According to the observations and ISORROPIA II simulated results,
particulate nitrate formation in Nanjing was $HNO_3$-limited, suggesting that the control
of NOx emissions will be able to decrease the nitrate concentration and improve the
air quality in this industrial city.
During the last decade, the mass ratios of nitrate-to-sulfate in $PM_{2.5}$ in the YRD
region have been found to range from 0.3 to 0.7 (Lai et al., 2007; Wang et al., 2003;
2006; Yang et al., 2005; Yao et al., 2002), reflecting that the $SO_4^{2-}$ concentration was
much higher than the $NO_3^-$ concentration. In the current study, the average mass ratio
of nitrate-to-sulfate was 1.1. Indeed, high nitrate-to-sulfate mass ratios of > 1 were
also observed in other mega-cities of China recently (Ge et al., 2017; Wei et al., 2018;
Ye et al., 2017; Zou et al., 2018). The elevated nitrate-to-sulfate ratio should be due to
the dramatic reduction of $SO_2$ emissions. The enhanced ratio also suggests that we
should pay more attention to develop some strategies for the reduction of NOx
emissions, leading to declined nitrate concentrations in the atmosphere and
improvement of the air quality in China.

**Data availability**
All the data used in this paper are available from the corresponding author upon
request (dryanlinzhang@outlook.com or zhangyanlin@nuist.edu.cn).

**Author contributions**
YLZ conceived and designed the study. YCL analyzed the data and wrote the
manuscript with YLZ. FM and MB performed aerosol sampling and data analyses
with YCL.

**Competing interests**
The authors declare that they have no conflict of interest.

**Acknowledgements**
This study was financially supported by the National Key R&D Program of China
(Grant No. 2017YFC0212700), the Natural Scientific Foundation of China (Nos.
41761144056, 91644103 and 41977185) and Jiangsu Innovation & Entrepreneurship
Team.

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

**Table Captions**
Table 1 The regression models between $NO_3^-/SO_4^{2-}$ (Y) and $NH_4^+/SO_4^{2-}$ (X) along
with the criterion values of $NH_4^+/SO_4^{2-}$ in ammonium-rich regime during the
sampling periods.

**Figure Captions**
Figure 1 Time series of concentrations in (a) $PM_{2.5}$ mass, (b) SIA species, (c) ALWC
and (d) Ox along with (e) Fn observed in Nanjing during the sampling
periods. The grey shadows represent the high $PM_{2.5}$ periods discussed in the
section 3.6.
Figure 2 Abundance of each species in TWSIIs during the (a) entire, (b) haze ($PM_{2.5} \geq$
150 $\mu g\ m^{-3}$ ) and (c) clear ($PM_{2.5} < 35\ \mu g\ m^{-3}$) events. The numbers in the
parentheses are standard deviations.
Figure 3 Abundance of each species in TWSIIs during the (a) entire, (b) haze ($PM_{2.5} \geq$
150 $\mu g\ m^{-3}$ ) and (c) clear ($PM_{2.5} < 35\ \mu g\ m^{-3}$) events. The numbers in the
parentheses are standard deviations.
Figure 4 The PSCF maps of high nitrate pollution.
Figure 5 Scatter plots of molar ratios of $NO_3^-/SO_4^{2-}$ against $NH_4^+/SO_4^{2-}$ in Nanjing
during the different seasons.
Figure 6 Scatter plot of $NO_3^-$ vs. excess-$NH_4^+$ molar concentrations in Nanjing during
the different seasons. The results in Beijing, Shanghai, Guangzhou, Lanzhou
and Hong Kong are also shown in this figure.
Figure 7 Scatter plots of (a) Fn against Ox and (b) Fn against ALWC in daytime and
nighttime aerosol samples during the high hourly $PM_{2.5}$ concentration
conditions (hourly $PM_{2.5} \geqq 150\ \mu g\ m^{-3}$).
Figure 8 Time series of concentrations in (a) $PM_{2.5}$ mass and CO, (b) SIA species
(NO$_3^-$, SO$_4^{2-}$ and NH$_4^+$), (c) ALWC, Ox and NO$_2$ and (d) RH and T in
Nanjing City from March 3 to 6, 2016. The grey shadows denote PM$_{2.5}$
episodes. The red numbers represent NO$_3^-$ production rate during the PM$_{2.5}$
episodes.
Figure 9   Nitrate concentrations simulated by ISORROPIA II model dependening on
TN and TA concentrations under (a) SO$_4^{2-}$ = 10 μg m$^{-3}$ and (b) SO$_4^{2-}$ = 60
μg m$^{-3}$. The purple dots denote the observed TN and TA concentrations at
the receptor site during the sampling periods.

Table 1 The regression models between $NO_3^-/SO_4^{2-}$ (Y) and $NH_4^+/SO_4^{2-}$ (X) along with the criterion values of $NH_4^+/SO_4^{2-}$ in ammonium-rich regime during the sampling periods.

| Sampling periods | Regression models | Criterion values of $NH_4^+/SO_4^{2-}$ |
|---|---|---|
| 2016 spring | $Y = 0.71 X – 1.27$; $R^2 = 0.87$ | 1.8 |
| 2016 summer | $Y = 0.67 X – 1.22$; $R^2 = 0.86$ | 1.8 |
| 2017 winter | $Y = 0.81 X – 1.50$; $R^2 = 0.91$ | 1.9 |
| 2017 spring | $Y = 0.95 X – 1.91$; $R^2 = 0.94$ | 2.0 |
| 2017 summer | $Y = 0.79 X – 1.32$; $R^2 = 0.84$ | 1.7 |

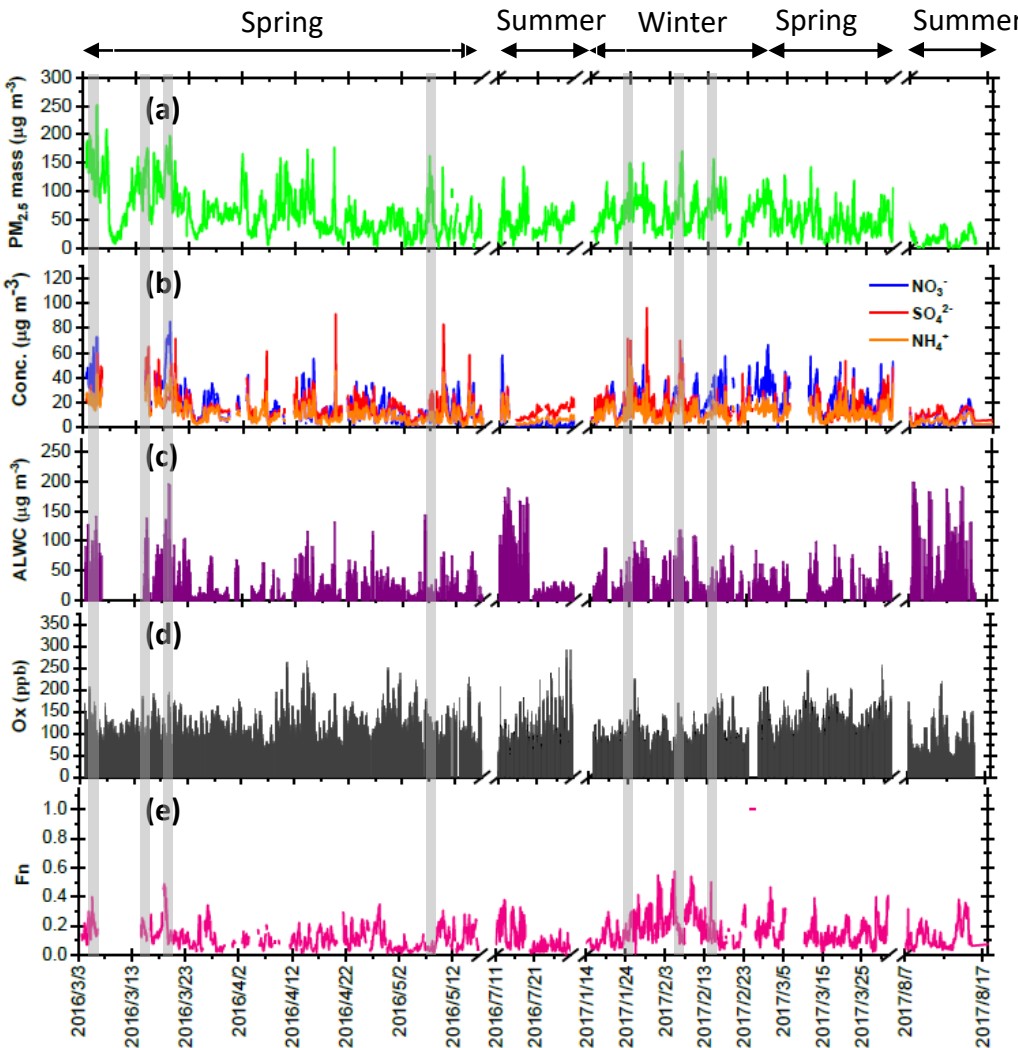

**Figure 1** Time series of concentrations in (a) PM$_{2.5}$ mass, (b) SIA species, (c) ALWC

and (d) Ox along with (e) Fn observed in Nanjing during the sampling periods.

The grey shadows represent the high PM$_{2.5}$ periods discussed in the section

3.6.

**(a) Entire days: PM$_{2.5}$ = 58 ± 35 μg m$^{-3}$**

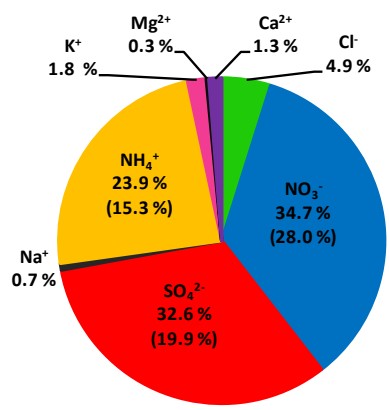

**(b) Haze events: PM$_{2.5}$ = 171 ± 18 μg m$^{-3}$**

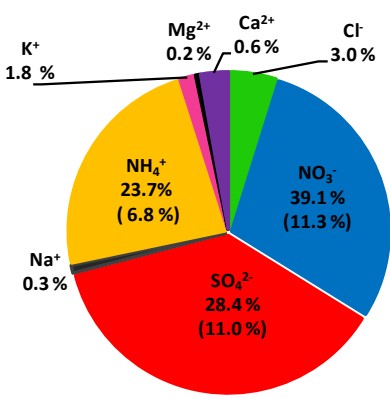

**(c) Clear events: PM$_{2.5}$ = 22 ± 9 μg m$^{-3}$**

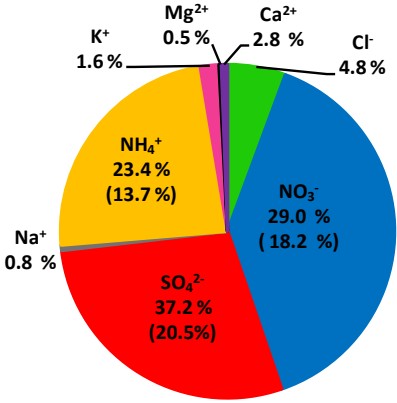

**Figurer 2** Abundance of each species in TWSIIs during the (a) entire, (b) haze (PM$_{2.5}$ ≥ 150 μg m$^{-3}$ ) and (c) clear (PM$_{2.5}$ < 35 μg m$^{-3}$) events. The numbers in the parentheses are standard deviations.

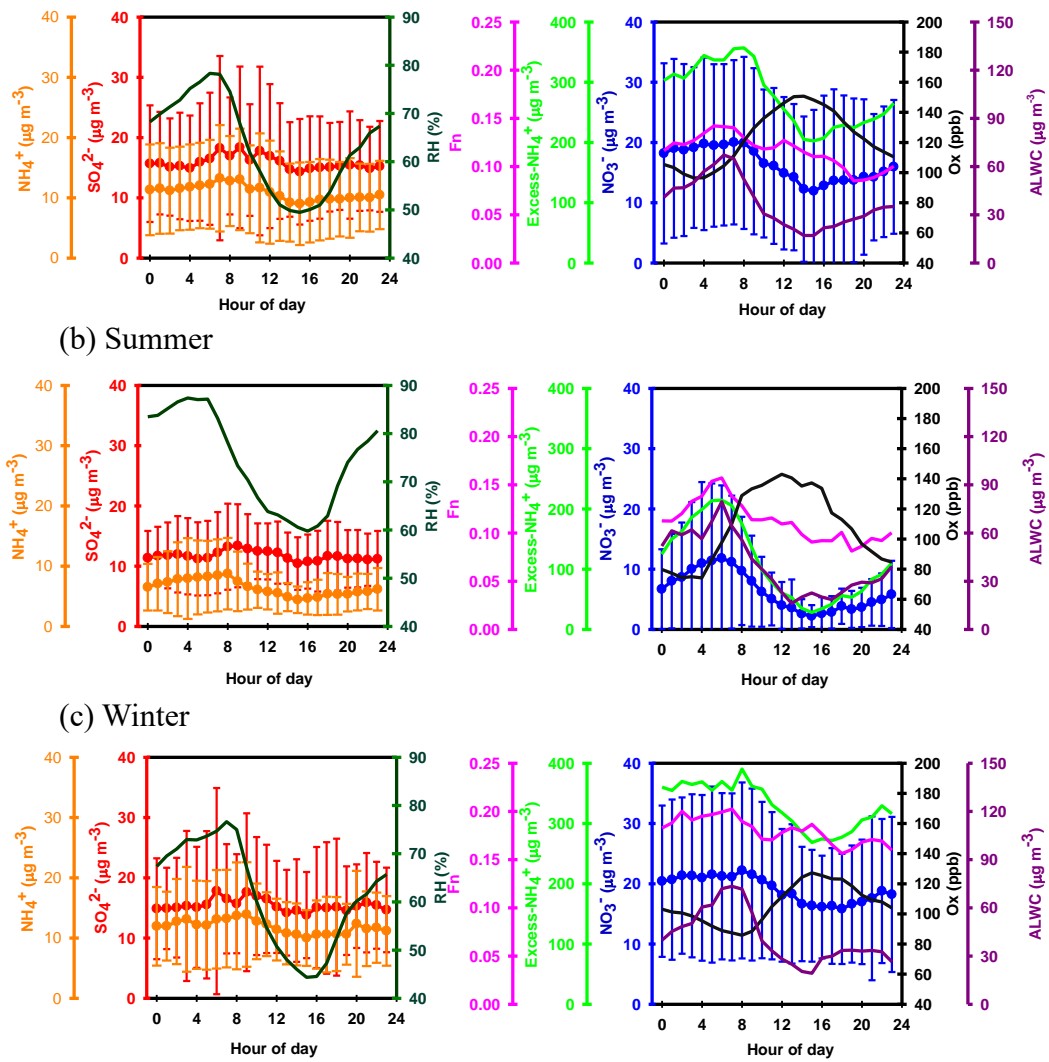

(b) Summer

(c) Winter

**Figure 3** Diurnal variations of the concentrations of $NO_3^-$, $SO_4^{2-}$ and $NH_4^+$, excess-$NH_4^+$, Ox and ALWC, and nitrogen conversion ratio (Fn) as well as ambient relative humidity in Nanjing during the sampling periods. For $SO_4^{2-}$, $NO_3^-$ and $NH_4^+$, the mean values (dots) and standard deviations (solid lines) are plotted.

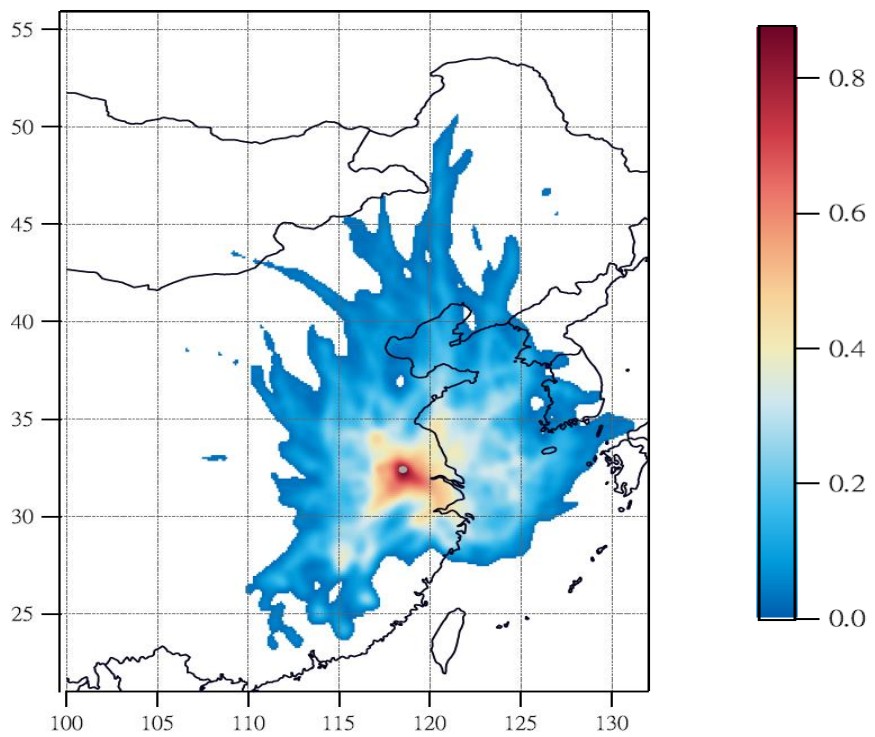

**Figure 4** The PSCF maps of high nitrate pollution.

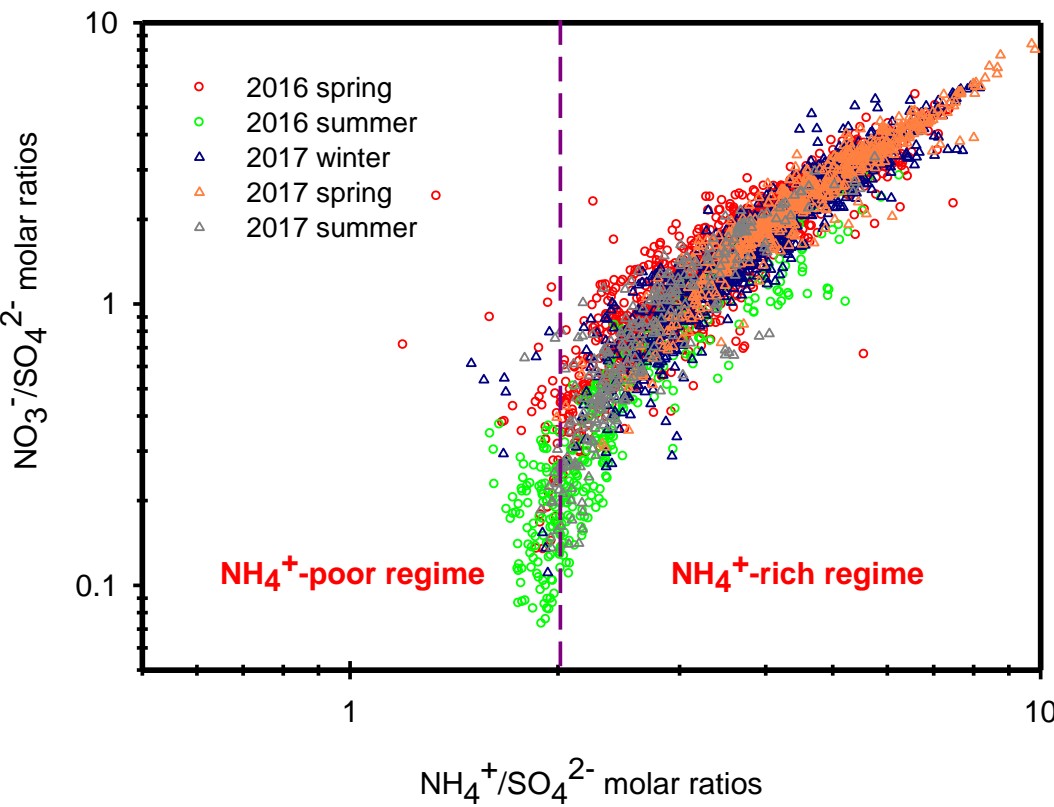

**Figure 5** Scatter plots of molar ratios of $NO_3^-/SO_4^{2-}$ against $NH_4^+/SO_4^{2-}$ in Nanjing

during the different seasons.

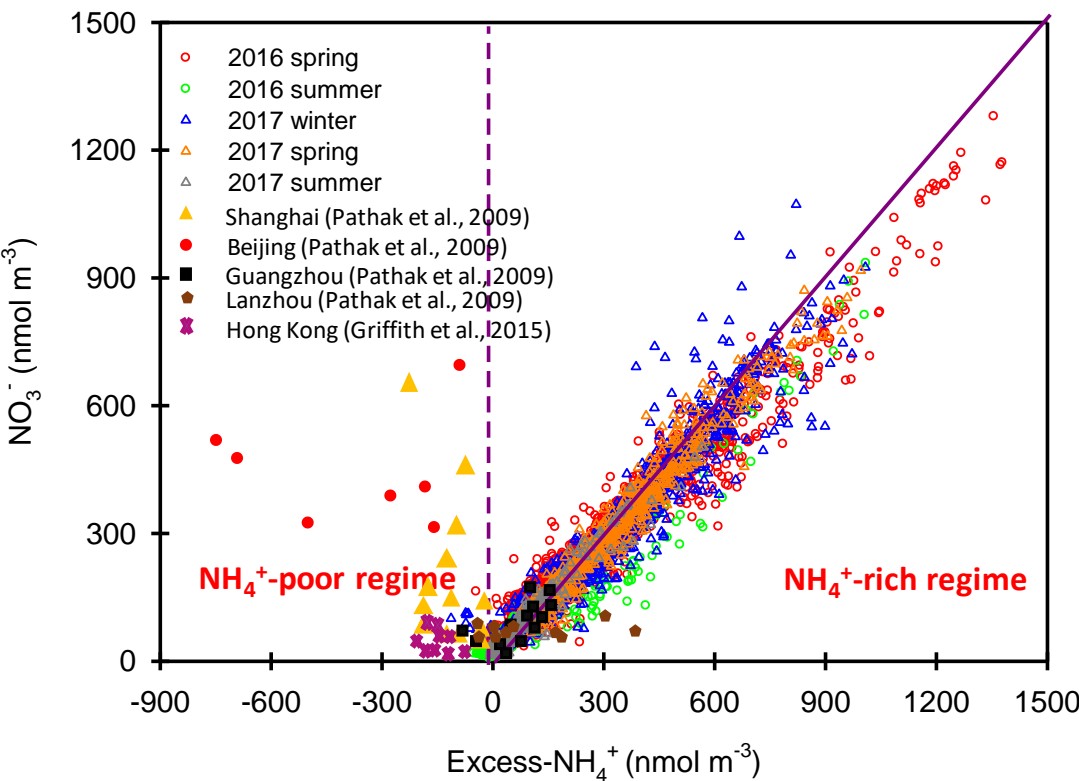

**Figure 6** Scatter plot of $NO_3^-$ vs. excess-$NH_4^+$ molar concentrations in Nanjing during

the different seasons. The results in Beijing, Shanghai, Guangzhou, Lanzhou

and Hong Kong are also shown in this figure.

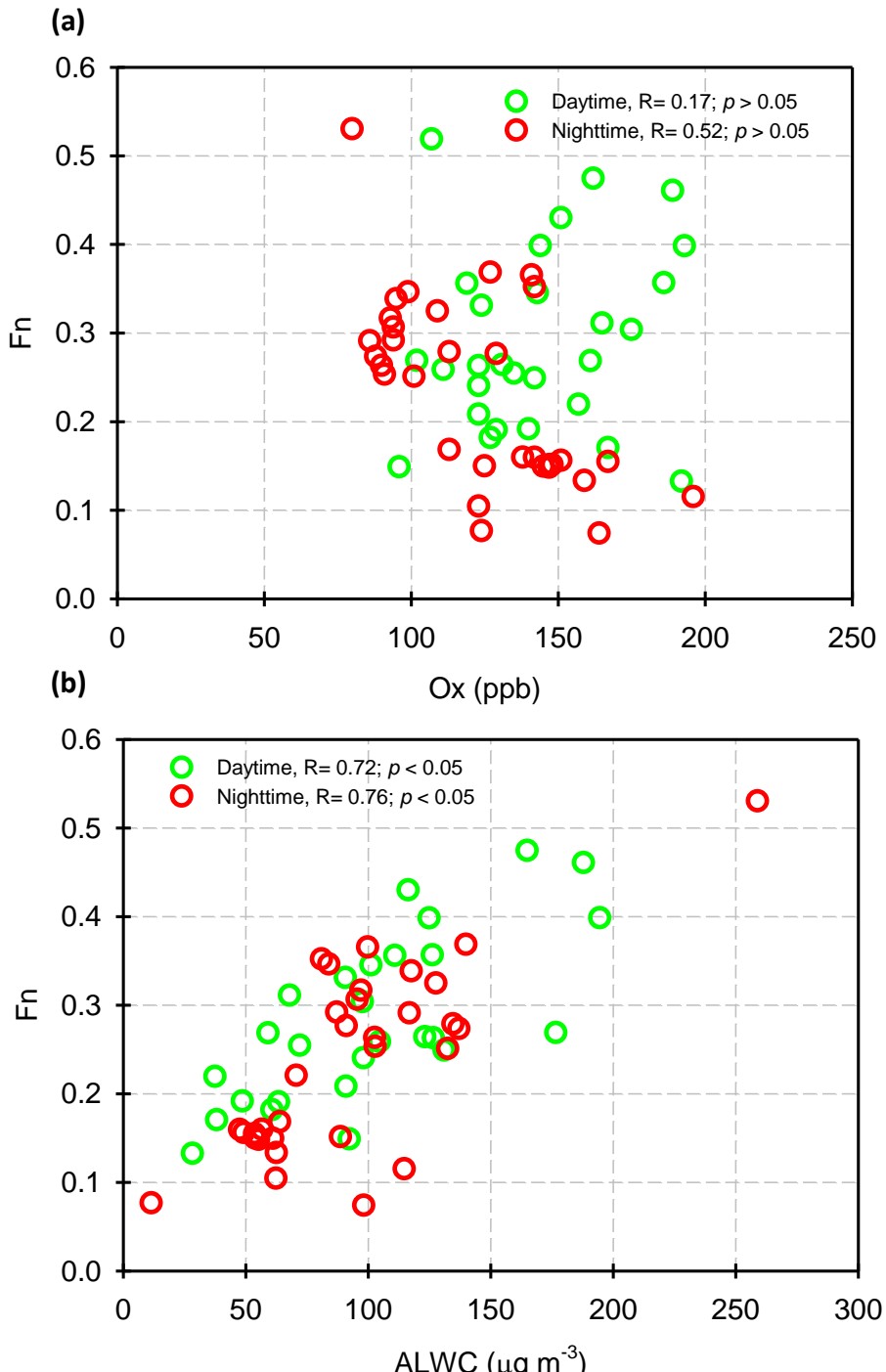

**Figure 7** Scatter plots of (a) Fn against Ox and (b) Fn against ALWC in daytime and

nighttime aerosol samples during the high hourly PM$_{2.5}$ concentration

conditions (hourly PM$_{2.5}$ $\geqq$ 150 μg m$^{-3}$).

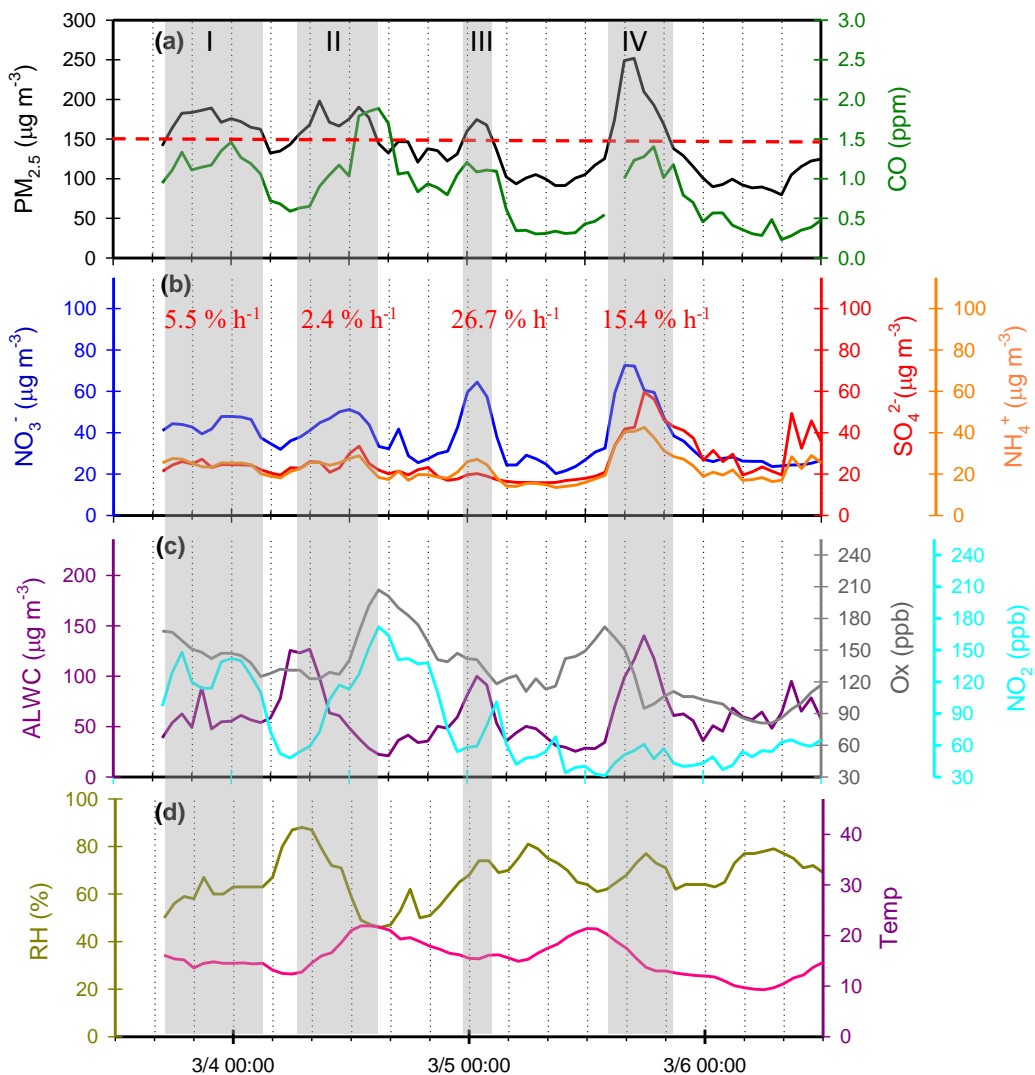

**Figure 8** Time series of concentrations in (a) PM$_{2.5}$ mass and CO, (b) SIA species

(NO$_3^-$, SO$_4^{2-}$ and NH$_4^+$), (c) ALWC, Ox and NO$_2$ and (d) RH and T in

Nanjing City from March 3 to 6, 2016. The grey shadows denote PM$_{2.5}$

episodes. The red numbers represent NO$_3^-$ production rate during the PM$_{2.5}$

episodes.

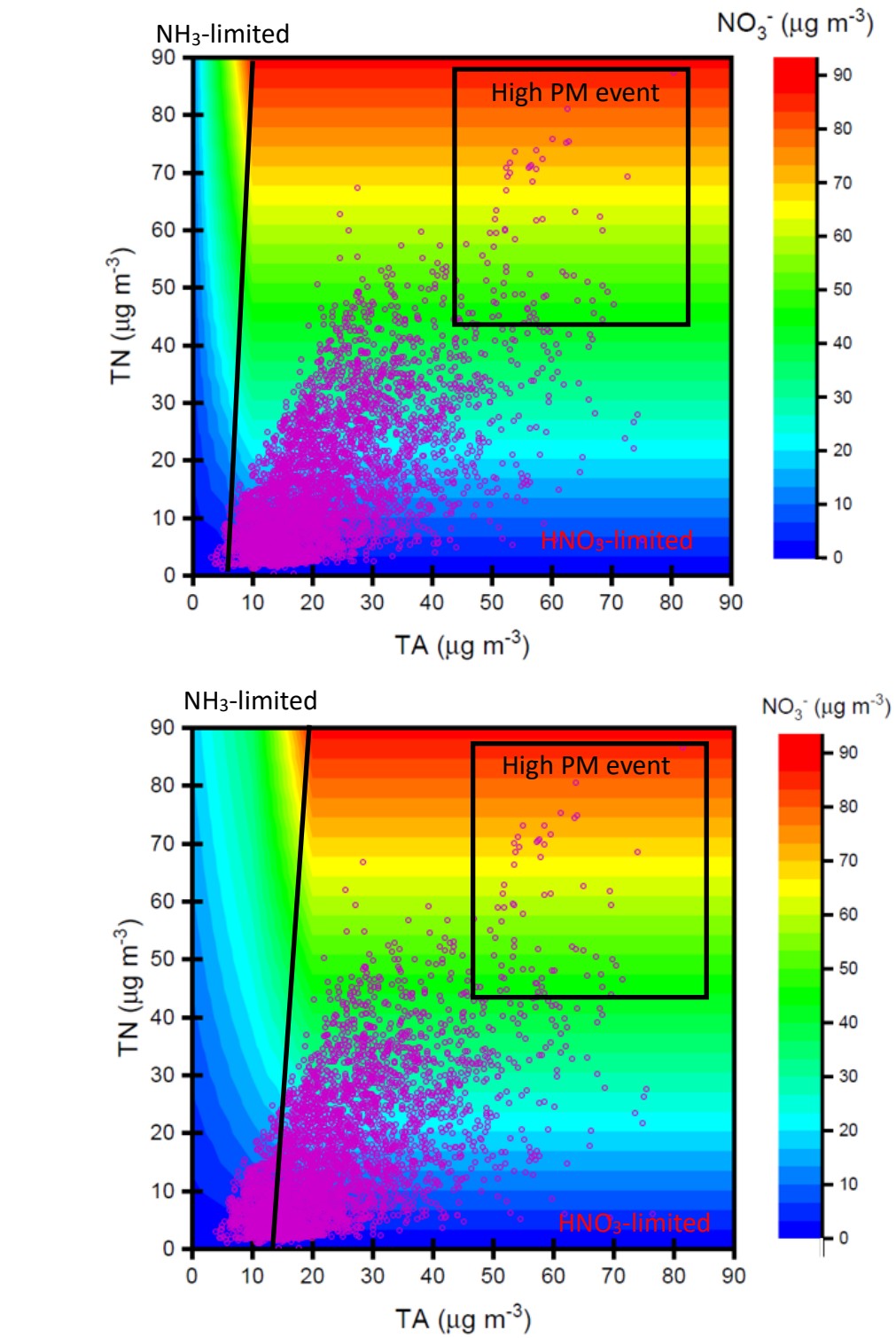

**Figure 9** Nitrate concentrations simulated by ISORROPIA II model depending on TN and TA concentrations under (a) $SO_4^{2-} = 10$ μg m$^{-3}$ and (b) $SO_4^{2-} = 60$ μg m$^{-3}$. The purple dots denote the observed TN and TA concentrations at the receptor site during the sampling periods.