# Peer review of "Heterogeneous Formation of Particulate Nitrate under Ammonium-rich Regimes during the High PM$_{2.5}$ Events in Nanjing, China"

_Atmospheric Chemistry and Physics, 2019_

## Referee Comment (RC1) · Anonymous Referee #1 · 8 Oct 2019

$SO_2$ has been significantly reduced in recent years; therefore, the nitrate aerosols become more and more important in China. This study presented a long-time measurement (2016-2017) of water-soluble ions of $PM_{2.5}$ in Yangtze River Delta (Nanjing), China. They found that the nitrate was a major faction of the $PM_{2.5}$ mass. From their study, they found the nitrate was mostly produced by the $N_2O_5$ heterogeneous uptake under the $NH_4^+$-rich condition. This study suggests the studied region is located in $HNO_3$-limit regime and thus the reduction of $NO_x$ may be more helpful to mitigate the PM pollution. The results could help the understanding of the heterogeneous formation of $NO_3$- aerosol in East China. This paper is well written, the method is sound, and the results could be important for aerosol chemistry community. I recommend this paper may be published after the following comments be addressed.

Line 48: reference is missing here.

Lines 130-136: QA/QC (ion balance) should be provided.

Line 209: should provide more evidence

Lines254-282: the criterion value should be explained in the very beginning.

Lines 328-333: are there any difference between day and night samples?

Lines 360-378: I suggest comparing NO3 increase rate with those reported in other studies

Lines 388-390: I think the authors should present more details about the Eq2. How to get this equation?

---

## Referee Comment (RC2) · Anonymous Referee #2 · 12 Nov 2019

This manuscript "Heterogeneous Formation of Particulate Nitrate under Ammonium-rich Regime during the high PM2.5 events in Nanjing, China" by Lin et al. investigates the formation of nitrate under polluted conditions. The conclusion on the importance of the heterogeneous formation of nitrate is drawn from the good linear relationship between the nitrogen conversion ratios (Fn) and aerosol liquid water (ALW), in contrast to the poor correlation between nitrate and Ox. However, good linear relationship does not necessarily correspond to a causal relationship between the two variables. This is particularly true for the semi-volatile nitrate, the partitioning of which between gas and particle-phases is also associated with other factors such as temperature and RH. The morning peak of nitrate as reported in this manuscript (Fig. 4) could be caused

by the photo-chemical production of nitrate after sun rise and the subsequent condensation on pre-existing particles before the temperature increases and RH decreases afterwards. Also, regional transport should be considered. In particular, the evaporation and re-condensation of nitrate during transport could also result in an increase of nitrate in the morning, coinciding with increase of ALW. Apart from the major issues mentioned above, there are several parts of the manuscript that need better interpretation or clarification as can be seen below.

Specific comments: Line 33-35. What is the absolute production rate? Please provide both ug m-3 h-1 and % h-1.

Line 146-148. It is not clear what are the inputs for the ISORROPIA II model. Why use "forward" instead of "backward" mode? One should acknowledge the uncertainty of this model. Specifically, is organics causing uncertainty in the model?

Line 179-180. Please provide correlation coefficient here.

Line 198-200. It is not clear how the theoretical equilibrium constants were derived? I saw large disagreement between the theoretical and observed values in every season in Fig. S3.

Line 266-268. Negative values of axcess NH4+ means deficit instead of excess. Please reword here.

Line 257-259. It is not clear how the criteria values were derived from the linear regression model. Is it the intercept or slope?

Line 302-306. Why not correct for background values for the calculation of Fn. Background values could have a big effect on Fn, leading to large uncertainties. Also, the diffusion rate or the distribution of gas and particles are different.

Line 310-312. Please explain why they are comparable, and why higher than other sites rather than just comparison of values.

Line 318-319. You are saying nitrate was totally formed from gas phase reaction during the day-time. Then, where does nitrate go during the night? It disappears or is transported to downwind sites? If transported, then, the nitrate transported form your upwind site could reach your sampling site in the evening?

Line 329-331. You mentioned that nitrate could be formed through gas-phase processes. Here, you are saying it is not from gas-phase reactions because of poor linear correlation between Fn and Ox.

Line 364-366. It is not clear how to get 70%. What are the absolute values of growth rate here, and in other cases?

Line 384-385. You are assuming HNO3+NH4 is the major pathway, but in previous part and in Table S1. Night time N2O5+H2O is the major pathway.

Figure 1. please provide better resolution. Also, the color bar has repeated 102, and Figure 1 is not discussed in the main text. If the goal of Fig. 1 is to show the location of the sampling site, it should be in supplementary.

Figure 2. Please provide year on the x-axis. Also indicate season and the cases you selected in Table S1 and Fig. 8.

---

## Author Comment (AC1) · 22 Dec 2019

The comment was uploaded in the form of a supplement: https://www.atmos-chem-phys-discuss.net/acp-2019-752/acp-2019-752-RC1-supplement.pdf

$SO_2$ has been significantly reduced in recent years; therefore, the nitrate aerosols become more and more important in China. This study presented a long-time measurement (2016-2017) of water-soluble ions of $PM_{2.5}$ in Yangtze River Delta (Nanjing), China. They found that the nitrate was a major faction of the $PM_{2.5}$ mass. From their study, they found the nitrate was mostly produced by the $N_2O_5$ heterogeneous uptake under the $NH_4^+$-rich condition. This study suggests the studied region is located in $HNO_3$-limit regime and thus the reduction of $NO_x$ may be more helpful to mitigate the PM pollution. The results could help the understanding of the heterogeneous formation of $NO_3$- aerosol in East China. This paper is well written, the method is sound, and the results could be important for aerosol chemistry community. I recommend this paper may be published after the following comments be addressed.

Line 48: reference is missing here.

Lines 130-136: QA/QC (ion balance) should be provided.

Line 209: should provide more evidence

Lines254-282: the criterion value should be explained in the very beginning.

Lines 328-333: are there any difference between day and night samples?

Lines 360-378: I suggest comparing NO3 increase rate with those reported in other studies

Lines 388-390:I think the authors should present more details about the Eq2. How to get this equation?

**Response to Reviewer's comments**

**(Manuscript No. ACP-2019-752)**

**Reviewer #1**

SO2 has been significantly reduced in recent years; therefore, the nitrate aerosols become more and more important in China. This study presented a long-time measurement (2016-2017) of water-soluble ions of PM2.5 in Yangtze River Delta (Nanjing), China. They found that the nitrate was a major faction of the PM2.5 mass. From their study, they found the nitrate was mostly produced by the N2O5 heterogeneous uptake under the NH4+-rich condition. This study suggests the studied region is located in HNO3-limit regime and thus the reduction of NOx may be more helpful to mitigate the PM pollution. The results could help the understanding of the heterogeneous formation of NO3- aerosol in East China. This paper is well written, the method is sound, and the results could be important for aerosol chemistry community. I recommend this paper may be published after the following comments be addressed.

**1st comment**

Line 48: reference is missing here.

> **Author's response:**
>
> As suggested, we have added a reference here (Huang et al., 2018) in the revised manuscript. (line 49 on page 2)

**2nd comment**

Lines 130-136: QA/QC (ion balance) should be provided.

> **Author's response:**
>
> The results of ion balance has been written in lines 205-211 and shown in Figure S2. Good correlations were found between cations and anions during the various sampling periods. The ratio of cation-to-anion was very close to 1.0 during each season, reflecting

good quality of our data in this study.

**3rd comment**

Line 209: should provide more evidence

**Author's response:**

Thanks for the reviewer's comment. Previous studies showed that the build-up of nitrate concentrations at the urban site in the early morning was due to enhanced nitrate formation in the residual layer in the mixing troposphere (Baasandorj et al., 2017, ES&T; Prabhakar et al., 2017, ACP). To explain this point, chemical model simulations are needed. However, in this work, we mainly focused on explaining the particulate nitrate behaviors based on observations and we cannot provide more evidence to support the argument as mentioned above. Thus, we have removed the sentence of "The higher nitrate……in the mixing troposphere (Baasandorj et al., 2017; Prabhakar et al., 2017)." in the revised manuscript.

**4th comment**

Lines254-282: the criterion value should be explained in the very beginning.

**Author's response:**

Thanks for the reviewer's comment. The criterion value can be calculated as the absolute value of intercept dividing by the slope in each linear regression model. (lines 309-310 on page 12)

**5th comment**

Lines 328-333: are there any difference between day and night samples?

**Author's response:**

Thanks for the reviewer's comment. As suggested, we made correlation analysis of Fn

vs. Ox and Fn vs. ALWC during the high PM$_{2.5}$ events for daytime and nighttime aerosol samples. The results showed that weak correlations between Fn and Ox were found in both daytime and nighttime. In contrast, Fn correlated very well with ALWC in both daytime and nighttime aerosol samples. This suggested that heterogeneous process played an important role in forming nitrate aerosols during both daytime and nighttime in the high PM$_{2.5}$ episodes. (line 388 on page 15, lines 389-394 on page 16 and Figure 7)

[Figure]

Figure 7 Scatter plots of (a) Fn against Ox and (b) Fn against ALWC in daytime and nighttime aerosol samples during the high hourly PM$_{2.5}$ concentration conditions (hourly PM$_{2.5}$ $\geqq$ 150 $\mu$g m$^{-3}$).

**6th comment**

Lines 360-378: I suggest comparing NO3 increase rate with those reported in other studies.

**Author's response:**

As suggested, we have compared the production rate of $NO_3^-$ between this work and the previous studies. In this work, the average $NO_3^-$ production rate due to heterogeneous process was $12.6 \pm 7.3$ % $h^{-1}$ ($4.1 \pm 3.6$ μg $m^{-3}$ $h^{-1}$). Previous studies showed that heterogeneous process of nitrate formation exhibited rates of $14.4$ % $h^{-1}$ (file measurement) and $14.3$ % $h^{-1}$ (lab. work). Our value was in accordance with those of the literatures (Calvert and Stockwell, 1983, ES&T; Pathak et al., 2011, ACP). On the contrary, the average growth rate of $NO_3^-$ by gas-oxidation process was $2.5 \pm 0.1$ % $h^{-1}$ ($0.8 \pm 0.3$ μg $m^{-3}$ $h^{-1}$). This value was in line with that ($2.4$ % $h^{-1}$) in the subtropical polluted urban site where nitrate aerosols were mainly produced by gas-oxidation reaction (Lin et al., 2007). Moreover, we also found some cases in which the elevated $NO_3^-$ might have been from both gas-phase and heterogeneous reactions, and the corresponding growth rate of $NO_3^-$ was approximately $7.5 \pm 3.0$ % $h^{-1}$ ($2.5 \pm 0.2$ μg $m^{-3}$ $h^{-1}$).(lines 432-440 on page 17 and lines 441-445 on page 18)

**7th comment**

Lines 388-390: I think the authors should present more details about the Eq2. How to get this equation?

**Author's response:**

Thanks for the reviewer's comment. In the revised manuscript, we have re-organized the section of "3.7 $NH_3/HNO_3$ limitation of nitrate aerosol formation". We used the ISORROPIA II model to evaluate whether control of $NH_3$ or $HNO_3$ ($NO_x$) is a better way to reduce particulate $NO_3^-$ concentrations in Nanjing. ISORROPIA II is a

thermodynamic equilibrium model which is built based on the $Na^+$ - $Cl^-$ - $Ca^{2+}$ - $K^+$ - $Mg^{2+}$ - $SO_4^{2-}$ - $NH_4^+$ - $NO_3^-$ - $H_2O$ aerosol system (lines 140-143 on page 6). The input of this model includes the concentrations of total ammonium ($NH_3$ + $NH_4^+$), total chloride ($HCl$ + $Cl^-$), $SO_4^{2-}$, $Na^+$, $K^+$, $Mg^{2+}$ and $Ca^{2+}$ along with ambient T and RH. In addition to ALWC and pH, ISORROPIA II model can also fit the observed SIA species very well (lines 462-466 on page 18 and Figure S7). Thus, we used this model to predict the concentrations of particulate nitrate under different total nitrate and ammonium conditions (lines 469-491 on page 19) and we deleted Eq. 2 in the revised manuscript.

[Figure]

Figure S7 Scatter plots of modeled results vs. observations of $NO_3^-$, $SO_4^{2-}$ and $NH_4^+$ in PM$_{2.5}$ in Nanjing during the sampling periods.

---

## Author Comment (AC3) · 22 Dec 2019

**Response to the reviewer's comment**

**(Manuscript No. ACP-2019-752)**

**Reviewer #2**

**1st comment**

This manuscript "Heterogeneous Formation of Particulate Nitrate under Ammonium rich Regime during the high PM2.5 events in Nanjing, China" by Lin et al. investigates the formation of nitrate under polluted conditions. The conclusion on the importance of the heterogeneous formation of nitrate is drawn from the good linear relationship between the nitrogen conversion ratios (Fn) and aerosol liquid water (ALW), in contrast to the poor correlation between nitrate and Ox. However, good linear relationship does not necessarily correspond to a causal relationship between the two variables. This is particularly true for the semi-volatile nitrate, the partitioning of which between gas and particle-phases is also associated with other factors such as temperature and RH. The morning peak of nitrate as reported in this manuscript (Fig. 4) could be caused by the photo-chemical production of nitrate after sun rise and the subsequent condensation on pre-existing particles before the temperature increases and RH decreases afterwards. Also, regional transport should be considered. In particular, the evaporation and re-condensation of nitrate during transport could also result in an increase of nitrate in the morning, coinciding with increase of ALW. Apart from the major issues mentioned above, there are several parts of the manuscript that need better interpretation or clarification as can be seen below.

**Author's response:**

Thanks for the reviewer's comment. In terms of this comment, we gave the responses as follows:

(1) Gas-phase oxidation (step1: $NO_{2(g)} + OH_{(g)} \rightarrow HNO_{3(g)}$, step2: $HNO_{3(g)} + NH_{3(g)} \rightarrow NH_4NO_{3(s,aq)}$) and heterogeneous reaction ($N_2O_{5(g)} + H_2O_{(g)} \rightarrow 2HNO_{3(aq)}$) are important

mechanisms of particulate nitrate formation. In gas-phase oxidation, the reaction rate constant of step 1 is $2.4 \times 10^{-11} (T/300)^{-1.3}$, which depends on ambient temperature (T) (Seinfeld and Pandis, 1998). Assuming the concentrations of $NO_2$ and OH are, respectively, 100 ppb and $10^6$ molecules $cm^{-3}$ (the typical values in the atmosphere in China), we then obtain the production rate of $HNO_3$ in a range of 4.5-5.2 ppb $h^{-1}$ as the T is between 0°C (273°K) and 30°C (303°K) (see the figures in this response). However, assuming $NO_2$ concentration is 100 ppb and T is 25°C, we can find that the $HNO_3$ formation rate increased from 0.9 ppb $h^{-1}$ (OH=$2\times10^5$ molecules $cm^{-3}$) to 23.8 ppb $h^{-1}$ (OH=$8\times10^6$ molecules $cm^{-3}$). Obviously, OH seems to be the major factor affecting on $HNO_3$ production rather than T in step 1 in gas-phase oxidation reaction. In step 2, the production of nitrate particles is very sensitive to T and RH (Lin et al., 2007). For heterogeneous process, ALWC is an important factor to induce formation of liquid $HNO_3$ (Brown and Stutz, 2012, Chem. Soc. Rev.).

Without modeling and isotope techniques, the correlation analysis between nitrogen conversion ratio (Fn or nitrogen oxidation ratio, NOR) and other parameters (Ox, RH and ALWC) were commonly used to discuss whether gas-oxidation or heterogeneous process is a possible formation mechanism of particulate nitrate (Kohder, 2002, AE; Jansen et al., 2014, AE; Quan et al., 2015, AE). Fn is defined as the conversion ratio of $NO_2$ to $HNO_3$ (total nitrate in the text). As mentioned above, the conversion from $NO_2$ to $HNO_3$ is mainly influenced by OH, and ALWC. Therefore, we employed correlation analysis between Fn and Ox (proxy of OH) and ALWC to study the potential pathways of nitrate production; the result showed that Fn correlated well with ALWC. Apart from correlation analysis, we also selected twelve $PM_{2.5}$ episode cases to explore the variations of the concentrations of nitrate, Ox and ALWC. We found most of enhanced nitrate coincided with increasing ALWC and decreasing Ox. Thus, we concluded that heterogeneous process might be a major formation mechanism of

nitrate formation in Nanjing during the high-PM$_{2.5}$ events (lines 382-388 on page 15, lines 389-394 on page 16, lines 399-414 on page 16, lines 415-440 on page 17 and lines 441-448 on page 18)..

(2)We agreed the reviewer's comment, that is, the partition of NH$_4$NO$_3$ (semi-volatile-nitrate) between gas- and particle-phase is very sensitive to T and RH. However, Fn means the formation ratio of NO$_2$ to total nitrate. In other words, Fn is used to evaluate the capability of NO$_2$ to HNO$_3$ prior forming nitrate particles. As mentioned above, Fn is mainly affected by OH and ALWC through different production processes. Consequently, we used the correlation analysis between Fn vs. Ox and Fn vs. ALWC to explore the formation mechanisms of HNO$_3$ (also reflected the pathway of nitrate aerosol formation).

(3)As our observed data, high nitrate concentrations were found between 3 a.m. and 7 a.m.. The high nitrate concentrations in the early morning might be caused by the nitrate formation via heterogeneous reaction in the dark, and gas-phase oxidation after sunrise and the subsequent condensation on pre-existing particles before the temperature increased and RH decreased afterwards. Moreover, the lower planet boundary layer (PBL) might be another reason for enhanced nitrate in the early morning. In the revised manuscript, we have rephrase the sentences in lines 239-244 on page 10.

(4) Regional transported air pollution would induce haze formation in China and is an important scientific issue. Nevertheless, it is very difficult to differentiate local and regional transported nitrate based on observations. The potential source contribution function (PSCF) is a method for identifying source region of air pollutants based on the HYSPLIT model. It is also commonly used to differentiate long-range transported air pollution from local air pollution based on observations combining with backward trajectories (Zhang et al., 2013, ACP; Hui et al., 2018, AE). The details of definition and calculation of PSCF are written in lines 158-179.. To compute PSCF, the 80

percentile value of nitrate concentration was selected as the "high nitrate pollution" threshold. The results showed that the hotspots (red color) of nitrate are shown in the vicinity of Nanjing, suggesting that high nitrate pollution at the receptor site might not be contributed by long-range transported air pollution, but was likely contributed by local emissions (lines 275-284 on page 11 and lines 285-287 on page 12). In this work, the influences of ambient T, RH and long-range transported nitrate on Fn can be neglected (mentioned above). Consequently, we still used the correlation analysis between Fn and Ox (ALWC), and the variations of concentrations in $NO_3^-$, $O_x$ and ALWC to investigate the formation mechanisms of particulate nitrate during the high $PM_{2.5}$ events.

[Figure]

Figure The $HNO_3$ production rate of gas-oxidation depending on temperature and OH concentrations.

[Figure]

Figure 4 The PSCF maps of high nitrate pollution.

Specific comments:

**2nd comment**

Line 33-35. What is the absolute production rate? Please provide both μg m$^{-3}$ h$^{-1}$ and % h-1.

> **Author's response:**
>
> As suggested, we have provided the production rate of particle nitrate in both units of μg m$^{-3}$ h$^{-1}$ and % h$^{-1}$ in the revised manuscript. During the sampling periods, a total of twelve high PM$_{2.5}$ events were found. Seven episodes suggested that heterogeneous processes were the major pathway of nitrate formation and two episodes indicated that gas-oxidation was the dominant formation mechanism of nitrate. On average, the production rate of nitrate aerosols via heterogeneous reaction was 12.6 ± 7.3 % h$^{-1}$ (4.1 ± 3.6 μg m$^{-3}$ h$^{-1}$), which was 5 times higher than that by gas-oxidation reaction (2.5 ± 0.1 % h$^{-1}$, 0.8 ± 0.3 μg m$^{-3}$ h$^{-1}$). (lines 34-36 on page 2, lines 448-450 on page 18 and Table S1)

**3rd comment**

Line 146-148. It is not clear what are the inputs for the ISORROPIA II model. Why use "forward" instead of "backward" mode? One should acknowledge the uncertainty of this model. Specifically, is organics causing uncertainty in the model?

> **Author's response:**
>
> Thanks for the reviewer's comment. ISORROPIA II is one of models to estimate aerosol liquid water content (ALWC) and has been widely used in many studies (Fountoukis and Nenes, 2007, ACP; Bian et al., 2014, ACP; Guo et al., 2015, ACP; Liu et al., 2017, GRL). The input of ISORROPIA II includes the concentrations of the observed total nitrate (HNO$_3$+NO$_3^-$), total ammonium (NH$_3$ + NH$_4^+$), total chloride (HCl + Cl$^-$), SO$_4^{2-}$, Na$^+$, K$^+$, Mg$^{2+}$ and Ca$^{2+}$ along with measured ambient T and RH.

Two modes, such as "forward" and "reverse", can be selected under the different input conditions. "forward" problem is always used when the quantity of ambient temperature and relative humidity along with the total concentrations (gas + aerosol) of $NH_4^+$, $Cl^-$, $NO_3^-$, and aerosol concentrations of $SO_4^{2-}$, $Na^+$, $Ca^{2+}$, $K^+$ and $Mg^+$ are known. On the contrary, ISORROPIA II also offers the ability to solve the "reverse problem", in which known quantities are the concentrations of $NO_3^-$, $SO_4^{2-}$, $Cl^-$, $Na^+$, $NH_4^+$, $K^+$, $Mg^{2+}$ and $Ca^{2+}$ in the aerosol phase together with ambient T and RH. In this work, we employed an online MAGAR instrument to monitored water-soluble inorganic ions in the aerosol phase and $SO_2$, $HNO_3$, $HCl$ and $NH_3$ concentrations. We also obtained the data of ambient T and RH. Therefore, we selected the "forward" mode in ISORROPIA II model; the observed concentrations of total nitrate ($HNO_3+NO_3^-$), total ammonium ($NH_3 + NH_4^+$), total chloride ($HCl + Cl^-$), $SO_4^{2-}$, $Na^+$, $K^+$, $Mg^{2+}$ and $Ca^{2+}$ along with measured ambient T and RH served as input of this model (lines 143-155 on page 6 and line 156 on page 7). The earlier study suggested that ISORROPIA II model can fit ALWC very well with the uncertainty of 13-20 % (Fountoukis and Nenes, 2007, ACP; Bian et al., 2014, ACP; Guo et al., 2015, ACP; Liu et al., 2017, ACP). Organic aerosols are not taken into account in ISORROPIA II model and some organic species contribute water content in aerosols. According to the results by Bougiatioti et al. (2016, ACP), ALWC would be underestimate about 28 % without considering organic aerosols. (lines 143-148 on page 6)

**4th comment**

Line 179-180. Please provide correlation coefficient here.

**Author's response:**

The correlation coefficients between cation and anion were in a range of 0.98-0.99. As suggested, we have provided correlation coefficient in the revised manuscript. (lines

**5th comment**

Line 198-200. It is not clear how the theoretical equilibrium constants were derived? I saw large disagreement between the theoretical and observed values in every season in Fig. S3.

**Author's response:**

Thanks for the reviewer's comment. The reaction of $HNO_3$ + $NH_3$ is one of the pathway to produce particulate $NH_4NO_3$. In the reaction of $HNO_{3(g)}$ + $NH_{3(g)} \rightarrow NH_4NO_{3(s, aq)}$, the theoretical equilibrium constant between gaseous $HNO_3$ and $NH_3$ and particle $NH_4NO_3$ ($k_2$ in the main text) can be calculated as (Mozurkewich, 1993, AE):

$$\ln(k_2) = 118.87 - \frac{24084}{T} - 6.025\ln(T)$$

where T is ambient temperature with a unit of °K. As the relative humidity (aw=RH/100) is greater than the deliquescence relative humidity of $NH_4NO_3$, the influence of RH on $k_2$ should be considered. Thus, $k_2$ should be replaced by $k_2'$, namely:

$$k_2' = (P_1 - P_2(1-a_w) + P_3(1-a_w)^2) \times (1-a_w) \times (1-a_w)^{1.75} \times k_2$$

where

$$\ln(P_1) = -135.94 + \frac{8763}{T} + 19.12\ln(T)$$

$$\ln(P_2) = -122.65 + \frac{9969}{T} + 16.22\ln(T)$$

$$\ln(P_3) = -182.61 + \frac{13875}{T} + 24.46\ln(T)$$

Using an appropriate expression for the temperature and relative humidity dependence of $NH_4NO_3$ thermodynamic properties, the equilibrium constant can be calculated at a specific temperature and relative humidity. Apart from theoretical equilibrium

constants, we also have observed equilibrium constants which were derived by the products of observed total nitrate ($HNO_3 + NO_3^-$) and total ammonium ($NH_3 + NH_4^+$). If the observed equilibrium constants were higher than the theoretical ones, indicating $NH_4NO_3$ would be produced (see in *supplementary S1),* otherwise the nitrate and ammonium should be partitioned into gas-phase. As shown in Figure S4b, most of the observed equilibrium constants were higher than the theoretical ones, suggesting that $NH_4NO_3$ was formed during the most sampling periods. However, the lower observed and higher theoretical equilibrium constants were found during the summer periods. This could partly explain the lower nitrate (or lower $NH_4NO_3$) concentrations during the hot seasons (lines 229-232 on page 9 and lines 233-235 on page 10).

**6th comment**

Line 266-268. Negative values of excess $NH_4+$ means deficit instead of excess. Please reword here.

**Author's response:**

As suggested, we have changed the sentence of "The results revealed that …………
and only 1 % of the excess-$NH_4^+$ data were lower than zero, reflecting…." to "The results revealed that ………… and only 1 % of data showed deficit-$NH_4^+$ conditions, reflecting….." in line 318 on page 13.

**7th comment**

Line 257-259. It is not clear how the criteria values were derived from the linear

regression model. Is it the intercept or slope?

**Author's response:**

Thanks for the reviewer's comment. The criteria value can be calculated as the absolute value of intercept dividing by the slope in each linear regression model (He et al., 2012).

**8th comment**

Line 302-306. Why not correct for background values for the calculation of Fn. Background values could have a big effect on Fn, leading to large uncertainties. Also, the diffusion rate or the distribution of gas and particles are different.

**Author's response:**

Thanks for the reviewer's comment. Actually, Fn means nitrogen conversion ratio during each aerosol sampling duration time. Thus, we used the concentrations of $NO_3^-$, $HNO_3$ and $NO_2$ obtained in each sample to calculate the Fn values.

**9th comment**

Line 310-312. Please explain why they are comparable, and why higher than other sites rather than just comparison of values.

**Author's response:**

Thanks for the reviewer's comment. In this work, the conversion ratio of nitrogen (Fn) was 0.14, which was comparable to that (0.17) in Taichung city of Taiwan where both gas-oxidation and heterogeneous reaction were dominant formation mechanisms of atmospheric $HNO_3$ (or $NO_3$-). However, our Fn value was 2.3 time higher than that (0.06) in Dokki, Egypt (Khoder, 2002, AE). The reason of significant discrepancy of Fn between this work and that in Dokki was not clearly understood, but it might be attributed to different formation processes of $HNO_3$. In Dokki, gas-phase oxidation was the dominant pathway of $HNO_3$ production while heterogeneous process (R3) played an important role in $HNO_3$ formation in addition to gas-phase oxidation in Nanjing, especially during the high-$PM_{2.5}$ events. The production rate of $HNO_3$ by heterogeneous process was much higher than that by gas-phase oxidation. Consequently, the Fn value

was much higher in this study (). .

**10th comment**

Line 318-319. You are saying nitrate was totally formed from gas phase reaction during the day-time. Then, where does nitrate go during the night? It disappears or is transported to downwind sites? If transported, then, the nitrate transported form your upwind site could reach your sampling site in the evening?

**Author's response:**

Thanks for the reviewer's comment. In this sentence, the total nitrate does not mean nitrate concentration, but means the concentrations of "total nitrate" ($HNO_3 + NO_3^-$). In the atmosphere, gas-phase oxidation ($NO_{2(g)} + OH_{(g)} \rightarrow HNO_{3(g)}$) and heterogeneous process ($N_2O_{5(g)} + H_2O_{(aq)} \rightarrow 2HNO_{3(aq)}$) are potential pathways of $HNO_3$ formation (Here, the $HNO_3$ means the nitric acid before forming nitrate and is also called total nitrate in this work). The product, $HNO_3$, would then reacts with $NH_3$ (or would be neutralized by $NH_4^+$) to produce $NH_4NO_3$ aerosols. In this work, we used correlation analysis between Fn (nitrogen conversion ratio), Ox and ALWC together with cases studies to discuss weather gas-phase oxidation or heterogeneous process was a major formation mechanism of particulate nitrate during the high $PM_{2.5}$ levels. The equation of Fn can be expressed as:

$$F_n = \frac{GNO_3^- + PNO_3^-}{GNO_3^- + PNO_3^- + NO_2}$$

As seen, the numerator of Fn is the concentrations of total nitrate. The denominator is the total $NO_2$ concentrations. The results showed that there were two peaks of Fn. In the first peak, the elevated Fn coincided with increasing ALWC, suggesting heterogeneous reaction since ALWC is one of the key parameters to accelerate the

transformation of $N_2O_5$ to liquid $HNO_3$. On the contrary, a second peak of Fn was found in the early afternoon when Ox (Ox = $NO_2$ + $O_3$, an index of the oxidation capacity) concentrations increased, but ALWC decreased. This suggested that the $HNO_3$ formation might be mainly associated with the gas-phase reaction of $NO_2$ + OH during the daytime. To avoid misunderstanding for the reader, we have changed "total nitrate " to "$HNO_3$" in line 379 on page 15 in the revised manuscript.

**11th comment**

Line 329-331. You mentioned that nitrate could be formed through gas-phase processes. Here, you are saying it is not from gas-phase reactions because of poor linear correlation between Fn and Ox.

**Author's response:**

Thanks for the reviewer's comment. Both gas-phase oxidation process and heterogeneous reaction are potential formation mechanisms of particulate nitrate. In this work, the good correlation between Fn and ALWC (R = 0.72 and 0.76, p <0.05) in the high $PM_{2.5}$ events. On the contrary, no correlation between Fn and Ox was found under the high $PM_{2.5}$ conditions. Thus, nitrate formation during the high-$PM_{2.5}$ events in Nanjing was likely attributed to heterogeneous reactions rather than gas-phase processes. (lines 388 on page 15 and lines 389-394 on page 16)

**12th comment**

Line 364-366. It is not clear how to get 70%. What are the absolute values of growth rate here, and in other cases?

**Author's comment**

During the sampling periods, a total of twelve high $PM_{2.5}$ events were found. The significant enhancements of $NO_3^-$ were observed during all the $PM_{2.5}$ episodes. Seven

episodes suggested that heterogeneous process ($N_2O_5 + H_2O$) might be a major pathway for nitrate formation since elevated $NO_3^-$ levels coincided with increasing AWLC and decreasing Ox (or Ox remaining at a constant level). In the heterogeneous-process cases, five of them (Case III, Case IX, Case X, Case XI and Case XII in Table S1) were observed during the nighttime (17:00 – 6:00 on the next day), suggesting that approximately 70 % heterogeneous reaction was observed in the dark (lines 428-432 on page 17). Moreover, the growth rate of nitrate of each high PM$_{2.5}$ case is listed in Table S1. As seen, the growth rate of nitrate was from 2.4 to 26.7 % h$^{-1}$. On average, the production rate of $NO_3^-$ (12.6 % h$^{-1}$) by heterogeneous processes was 5 times higher than that (2.5 % h$^{-1}$) of gas-oxidation reactions. This might explain the abrupt increase of nitrate concentrations during the high PM$_{2.5}$ events (lines 448-451 on page 18).

**13th comment**

Line384-385. You are assuming $HNO_3+NH_3$ is the major pathway, but in previous part and in Table S1. Night time $N_2O_5+H_2O$ is the major pathway.

**Author's comment**

Thanks for the reviewer's comment. In the revised manuscript, we have re-organized the section of "3.7 NH$_3$/HNO$_3$ limitation of nitrate aerosol formation". We used the ISORROPIA II model to evaluate whether control NH$_3$ or HNO$_3$ (NO$_x$) is a better way to reduce particulate $NO_3^-$ concentrations in Nanjing since ISORROPIA II model can predict the concentrations of $NO_3^-$, $SO_4^{2-}$ and $NH_4^+$ very well (R$^2$=0.97-0.99 with all slopes of approximately 1.0) under thermodynamic equilibrium in aerosol system (lines 462-466 on page 18, lines 467-468 on page 19 and Figure S7). In this work, we used this model to calculate concentrations of particulate nitrate depending on various total nitrate and total ammonium concentrations under high- and low-sulfate concentrations.

The results showed that nitrate aerosol formation in Nanjing during the high PM$_{2.5}$ events was HNO$_3$-limited. This also reflected that control NOx emissions can reduce particulate nitrate concentrations.

[Figure]

Figure S7 Scatter plots of simulated concentrations of NO$_3^-$, SO$_4^{2-}$ and NH$_4^+$ against observed ones in Nanjing during the sampling periods.

[Figure]

Figure 9 Nitrate concentrations simulated by ISORROPIA II model depending on TA and TN concentrations under (a) SO$_4$2$^-$ = 10 μg m$^{-3}$ and (b) SO$_4^{2-}$ = 60 μg m$^{-3}$.

**14th comment**

Figure 1. please provide better resolution. Also, the color bar has repeated 102, and Figure 1 is not discussed in the main text. If the goal of Fig. 1 is to show the location of the sampling site, it should be in supplementary.

**Author's response:**

Thanks for the reviewer's comment. In the revised manuscript, we have replaced the new figure with a better resolution. However, this figure only shows the relative location of the sampling site and therefore we move this figure to the supplementary (Figure S1).

**15th comment**

Figure 2. Please provide year on the x-axis. Also indicate season and the cases you selected in Table S1 and Fig. 8.

**Author's response:**

As suggested, we have provide year on the x-axis in "new Figure 1" . Also, we have indicated the season and used shadows to pointed out the cases we selected in Table S1 and Fig. 8 in the revised manuscript.